# The most extreme rainfall erosivity event ever recorded in China up to 2022: The "7.20" storm in Henan province

Yuanyuan Xiao[1], Shuiqing Yin[1], Bofu Yu[2], Conghui Fan[1], Wenting Wang[1], Yun Xie[1]

[1]State Key Laboratory of Earth Surface Processes and Resource Ecology, Faculty of Geographical Science, Beijing Normal University, Beijing, 100875, China
[2]Australian Rivers Institute, School of Engineering and Built Environment, Griffith University, Nathan, Queensland, QLD 4111, Australia

*Correspondence to*: Shuiqing Yin (yinshuiqing@bnu.edu.cn)

**Abstract.** Severe water erosion occurs during extreme storm events. Such an exceedingly severe storm occurred in Zhengzhou in central China on 20 July 2021 (the "7.20" storm). The magnitude and frequency of occurrence of this storm event were examined in terms of how erosive it was. To contextualize this extreme event, hourly rainfall data from 2420 automatic meteorological stations in China from 1951 to 2021 were analyzed to: (1) characterize the spatial and temporal distribution of rainfall amount and rainfall erosivity of the "7.20" storm, (2) evaluate the average recurrence interval of the maximum daily and event rainfall erosivity, and (3) establish the geographical distribution of the maximum daily and event rainfall erosivity in China. The center of the "7.20" storm moved from southeast to northwest in Henan province, and the most intense period of rainfall occurred in the middle and late stages of the storm. Zhengzhou meteorological station happened to be aligned with the center of the storm, with a maximum daily rainfall of 552.5 mm and a maximum hourly rainfall intensity of 201.9 mm·h⁻¹. The average recurrence intervals of the maximum daily rainfall erosivity (43,354 ± 1863 MJ·mm·ha⁻¹·h⁻¹) and the maximum event rainfall erosivity (58,874 ± 2351 MJ·mm·ha⁻¹·h⁻¹) were estimated to be about 19,200 and 53,700 years, respectively, assuming the log Pearson type III distribution, and these were the maximum rainfall erosivity ever recorded among 2420 meteorological stations in mainland China up to 2022. The "7.20" storm suggests that the most erosive of storms does not necessarily occur in the wettest places in southern China, and it can occur in mid-latitude around 35 °N with a moderate mean annual rainfall of 566.7 mm in Zhengzhou.

**Keywords.** soil erosion, extreme rainfall, rainfall erosivity, the return period

## 1 Introduction

Soil erosion is a land degradation process that can affect food production, biodiversity, carbon stocks and ecosystem services (Kebede et al. 2021; Panagos et al., 2015). Soil erosion models are powerful tools to evaluate the rate of erosion and the effect of soil and water conservation measures for decision makers. The Universal Soil Loss Equation (USLE) (Wischmeier and Smith, 1965, 1978) and the revised USLE (Renard et al., 1997; USDA–ARS, 2013), and the Chinese Soil Loss Equation (CSLE, Liu et al., 2002) are widely used empirical soil erosion prediction models for estimating the long-term average amount of soil loss. Rainfall erosivity quantifies the potential ability of rainfall and runoff to erode the soil and represents the climatic effect on soil erosion as one of the factors in the USLE, RUSLE and CSLE (Yin et al., 2017).

Most studies have focused on the long-term average of rainfall and rainfall erosivity characteristics (Gu et al., 2020; Li et al., 2008; Liu et al. 2018; Yin et al., 2019), and have assessed the intensity and frequency of extreme rainfall events at the regional, national and global scales (Alexander et al., 2007; Almagro et al., 2017; Evans et al., 2016; Nearing et al., 2004). The long-term average value cannot fully represent the severity of the soil erosion process, and a few severe soil erosion events can contribute a great deal to the total amount of soil lost over many years (Bezak et al. 2021; Borrelli et al., 2016; Meusburger et al., 2012; Petek et al., 2018). For example, field observations at the plot scale in eastern Austria showed that the three largest

erosion events from 1994 to 2019 accounted for 79% of the total soil loss over the same period (Klik and Rosner, 2020). Zhou et al. (1992) reported that high-intensity, short-duration heavy precipitation events accounted for about 90% of the total annual soil erosion in the Loess Plateau region.

Extreme rainfall, which varies a great deal in space and time, can lead to severe flooding, with far-reaching implications for socio-economic and human activities (Fishman, 2016). With global warming, the frequency and intensity of extreme precipitation events are increasing mostly in mid-latitudes (Fang et al., 2017; IPCC 2021; Liao et al., 2019; Liu et al., 2017). Extreme rainfall, especially rainfall events with high intensity, is often more erosive (Fang et al., 2018; Huang et al., 2016a, 2016b, 2016c). Many studies reported that satellite-based products tended to underestimate the extreme rainfall, which can have an important effect on the estimation of rainfall erosivity using satellite-based products (Jiang et al., 2019; Palharini et al., 2020; Rahmawati and Lubczynski., 2018). For example, Bezak et al. (2022) showed CMORPH estimates had a marked tendency to underestimate rainfall erosivity in highly erosive areas when compared to the the GloREDa estimates. In addition, underestimation of extreme rainfall from climate models will lead to conservative projections of erosivity in highly erosive areas in the future (Panagos et al., 2022). Therefore, it is of great interest to examine the magnitude and frequency of occurrence of rainfall and rainfall erosivity of extreme storm events.

An extraordinarily heavy rainfall event occurred between 17th and 22th of July 2021 in Henan province. Such a rare event was never experienced or recorded up to 2022 in China. Record daily rainfall was observed at 10 meteorological stations in Zhengzhou, Xinxiang, Kaifeng, Zhoukou, Luoyang and other cities in Henan province. Zhang et al. (2021) reported that the storm was influenced by several weather systems including the eastward extension of the South Asian high, the abnormal northerly subtropical high, the Bengal Bay Depression at low latitude, the typhoon "Chapaca" in the South China Sea and the typhoon "Fireworks" in the Western Pacific. The strengthened and eastward extension of the South Asia high leads to an obvious divergence area of the upper atmosphere over Henan province, which is conducive to the upward movement of the lower atmosphere. The subtropical high, which is northward moving and stronger than usual for the same period, the No. 6 typhoon "Fireworks" and the No. 7 typhoon "Chapaca" in low latitudes, and the low pressure in Bengal Bay have led to the stable and lasting transmission of warm and humid airflow to Henan province (Zhang et al., 2021; Qian et al., 2022). Taihang Mountains and Funiu Mountains in the northwestern and western Henan province blocked the airflow, and a strong convergence formed in front of mountains, resulting in this extreme rainfall event.

The maximum hourly rainfall between 16:00 and 17:00 on 20 July reached 201.9 mm at Zhengzhou meteorological station, the highest ever recorded in China up to 2022 (Zhang et al., 2021). It has been widely reported that this extreme storm caused extensive flooding and landslides with damages to infrastructure and loss of human lives (Jin et al., 2022; Zhang et al., 2022). Event total rainfall, daily and hourly rainfall of the "7.20" storm have been reported elsewhere (Zhang et al. 2021), whereas rainfall erosivity associated with this extreme storm has not. The "7.20" storm presents a rare opportunity to examine the extreme rainfall erosivity in China. For this study, hourly rainfall data were used to evaluate the maximum daily and event rainfall erosivity, to estimate its average recurrence interval, to contextualize geographically the extreme erosivity of the "7.20" storm, to demonstrate how extreme the erosivity value of the "7.20" storm was and how large event rainfall erosivity could be in China, and to highlight the need to pay attention to extreme storm events and the huge erosion risk associated with them in the future.

## 2 Material and Methods

### 2.1 Data source and pre-processing

Observed hourly rainfall data from 1951 to 2021 for 2420 meteorological stations in China were collected by siphon rain gauges or tipping bucket rain gauges. The instrument used by China Meteorological Administration (CMA) is SL3-1 tipping

bucket rain sensors, and precipitation was measured according to the operation manual at all stations. Tipping bucket rain gauges have a rainfall bearing diameter of 200 mm, and its resolution is 0.1 mm. The maximum allowable rainfall intensity is 4 mm·min$^{-1}$, and the maximum allowable rainfall error is ± 4 mm for every 100 mm. A multi-sensor system was used for precipitation measurement. The system consists of three separate SL3-1 tipping bucket rain sensors. Multi-sensor automatic weather stations detect abnormal or missing rainfall data caused by rain sensor failures to ensure precipitation data quality (He and Huang, 2015). The rainfall data was acquired from CMA and the data had been quality-controlled by CMA's National Meteorological Information Center. However, we found some outlier in the data, so we checked hourly with daily observation from rain gauges. Hourly observations in early days were mainly digitized from precipitation autographic charts on paper. From 2000 to 2005, automatic weather stations were put into use and their introduction was gradually accelerated. Since 2005, nearly all observations were recorded with automatic weather stations. Hourly rainfall data from 796 meteorological stations in Henan and its surrounding nine provinces (municipalities) from 20:00 (Beijing time) on 16 July and to 20:00 on 22 July 2021 were used to characterize the "7.20" storm. Hourly rainfall data from 1951 to 2020 were used to calculate the annual maximum daily and event rainfall erosivity. To reduce the impact of missing values on the result, years with missing data were discarded. A year with missing data was defined as follows: if there were four or more hours of missing records on a given day, it was considered as a missing day and if the number of missing days in a month ≥ six, it was considered as a missing month. Since most of the rainfall in the north of China (north of 32°N) is concentrated from May to September, the year with any month from May to September missing was defined as a missing year. In southern China (south of 32°N), the year with any month from April to October missing is defined as a missing year. Missing years were removed, and missing values in effective years are input as zero value.

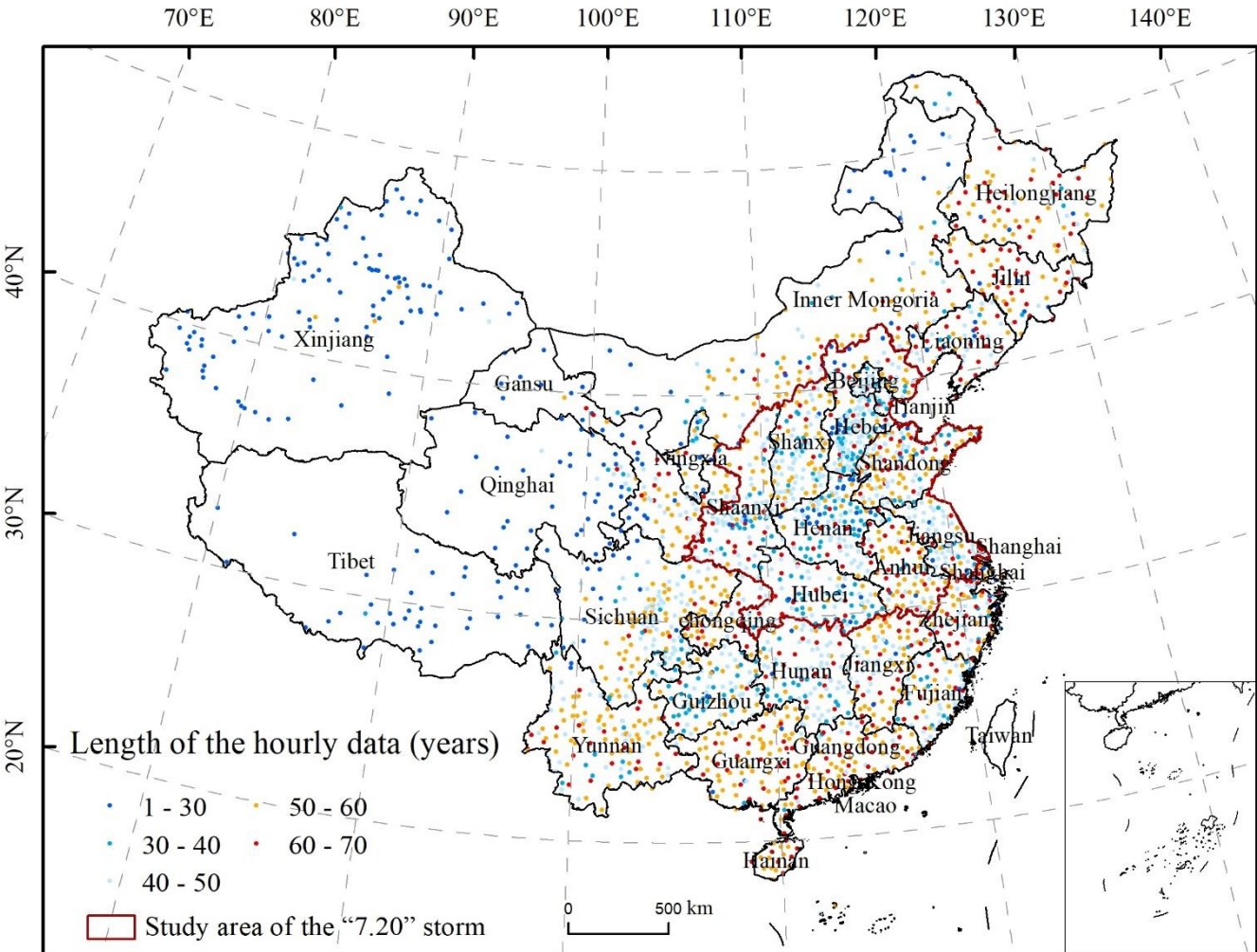

**Figure 1.** Spatial distribution of stations with hourly rainfall data and the record length.

 **2.2 Framework of study**

*2.2.1 Definition of rainfall events and rainfall parameters*

An event was defined as a period of rainfall separated with dry periods greater than "minimum inter-event time" (MIT). The MIT in the USLE and RUSLE2 is six hours. In this study, MIT of six hours was used to define rainfall events. The maximum event rainfall, maximum daily rainfall, maximum hourly rainfall and maximum event rainfall erosivity were computed
following to the framework shown in Fig. 2. Since there were multiple events over the six-day in period during the "7.20" storm, the maximum event rainfall was the maximum rainfall amount of all events over the six-days period. Maximum event rainfall erosivity was similarly defined.

*2.2.2 Calculation of the energy and daily/event rainfall erosivity*

Hourly data were used to calculate the rainfall erosivity, $EI_{30}$ (MJ·mm·ha$^{-1}$·h$^{-1}$) for each event, which is the product of the event
energy and peak 30-min intensity. All the hourly data for day (8:00 pm to 8:00 pm) were used to compute daily rainfall erosivity. Rainfall kinetic energy is used by most erosion models for assessing the capacity of rainfall to produce erosion. Rainfall kinetic energy is a function of raindrop size and falling velocity. Because the direct measurement of kinetic energy (KE) requires complex and expensive instruments, many different estimation methods have been developed. These methods use logarithmic, exponential, or power law formulas to derive kinetic energy-intensity (KE-I) relationships. The most widely accepted kinetic
energy-intensity relationship is the exponential model proposed by Kinnell (1981). This equation has the general form:

$$e_r = e_{max} \cdot [1 - a \cdot exp(-b \cdot i_r)] \tag{1}$$

where $e_{max}$, $a$, $b$ are empirical constants. Among them, the coefficients $a$ and $e_{max}$ determine the minimum kinetic energy content. On the other hand, the coefficient $b$ defines the general shape of the curve (Kinnell, 1981).

The rainfall kinetic energy is calculated by Eq. (3), which includes the modification suggested by McGregor et al. (1995).
The total energy (EN, MJ·ha$^{-1}$) of an erosive event was computed using following equations (USDA-ARS, 2013):

$$EN = \sum_{r=1}^{l}(e_r \cdot P_r) \tag{2}$$

$$e_r = 0.29 \cdot [1 - 0.72 \cdot exp(-0.082 \cdot i_r)] \tag{3}$$

where a rainfall event is divided into $l$ periods, each with an intensity, $i_r$ (mm·h$^{-1}$), $P_r$ (mm) is the rainfall amount for the $r$th period and $e_r$ (MJ·mm$^{-1}$·ha$^{-1}$) is the energy per unit rainfall per unit area for the $r$th period.

The event rainfall erosivity can be estimated with of $EN$ and $I_{1h}$ (USDA-ARS, 2013):

$$EI_{1h} = EN \cdot I_{1h} \tag{4}$$

$$EI_{30} = c \cdot EI_{1h} \tag{5}$$

where $I_{1h}$ is the peak one-hour rainfall intensity for the erosive event, c is the calibrated conversion factor for the rainfall erosivity from one-hour data to one-min data. Yue et al. (2020) used hourly rainfall data to calculate a conversion coefficient
of 1.489 for the 1-in-10-year EI$_{30}$, which is suitable for evaluating extreme rainfall erosivity on average. The conversion factor for individual stations in China ranged from 1.321 to 4.601, and the conversion factor for Zhengzhou Meteorological Station was 2.029, higher than the average, or expected, conversion factor used for this study (Fig. S2.1). We have included the standard error ± 0.064 for the conversion factor to indicate the likely uncertainty associated with this conversion factor.

Total rainfall and energy over the six days of the "7.20" storm for 796 stations were interpolated spatially at 100 m spatial
resolution, and the regional averages for Henan province and the study area (Henan province and its surrounding nine provinces/municipalities) were calculated and compared with Zhengzhou meteorological station.

The rainfall and rainfall erosivity maps were generated by the interpolation of rainfall and rainfall erosivity values from at-site rainfall observations by geostatistics techniques, such as the inverse distance weighting, or ordinal Kriging (Panagos et al., 2015; Sadeghi et al., 2017; Yin et al., 2019). We used inverse distance weighting (IDW) to interpolate point data to map rainfall

and rainfall erosivity distribution for the region. The IDW method computes precipitation at the interpolating point by assigning larger weights to observation stations closer to the target grid (Shepard, 1968).

### 2.2.3 Log Pearson type III distribution

An annual series is defined here as a collection of maxima, one from each calendar year. Annual series of the maximum daily and event rainfall erosivity from the period 1951-2020 ($n = 67$ due to three missing years) for Zhengzhou meteorological
station were sorted in a descending order with the largest assigned a rank of one. The empirical return period, or the average recurrence interval, of each observation in the annual series was calculated according to the following formula (Bobeé and Robitaille, 1977):

$$RP = \frac{n+1}{m} \tag{6}$$

where $RP$ is the empirical return period in years, $n$ is the number of years or the sample size, and $m$ is the rank ($m = 1$ for
the largest).

The probability distribution used to fit the annual series was the log Pearson type III (LP-III) distribution. LP-III distribution is considered a suitable model for flood frequency estimation in many investigations (Bobeé and Robitaille, 1977; England et al., 2003; England, 2019). The logarithms of the annual series of the maximum daily rainfall erosivity and the maximum event rainfall erosivity from 1951-2020 for Zhengzhou meteorological station were used to fit the Pearson type III distribution (P-
III), respectively. The probability density function (PDF) and cumulative distribution function (CDF) of P-III distribution model are as follows:

$$f(x) = \frac{\beta^\alpha}{\Gamma(\alpha)}(x - a_0)^{\alpha-1}e^{-\beta(x-a_0)} \quad x > a_0, \alpha > 0, \beta > 0 \tag{7}$$

$$F(x) = \frac{\beta^\alpha}{\Gamma(\alpha)}\int_{a_0}^{x}(x - a_0)^{\alpha-1}e^{-\beta(x-a_0)}dx \tag{8}$$

where x is the random variable of interest, $a_0$ is the location parameter, $\alpha$ the shape parameter, and $\beta$ the scale parameter.
$\Gamma(\alpha)$ is the gamma function. The basic parameters, mean $\bar{x}$, coefficient of variation $C_V$, and skewness coefficient $C_S$, were used to estimate parameter $a_0$, $\alpha$ and $\beta$ (Viessman Jr. and Lewis, 2002) (Eq. 9-11), and the 95% confidence interval was also estimated (Kite, 1975).

$$\alpha = \frac{4}{C_s} \tag{9}$$

$$\beta = \frac{2}{\bar{x}C_sC_v} \tag{10}$$

$$a_0 = \bar{x}(1 - \frac{2C_v}{C_s}) \tag{11}$$

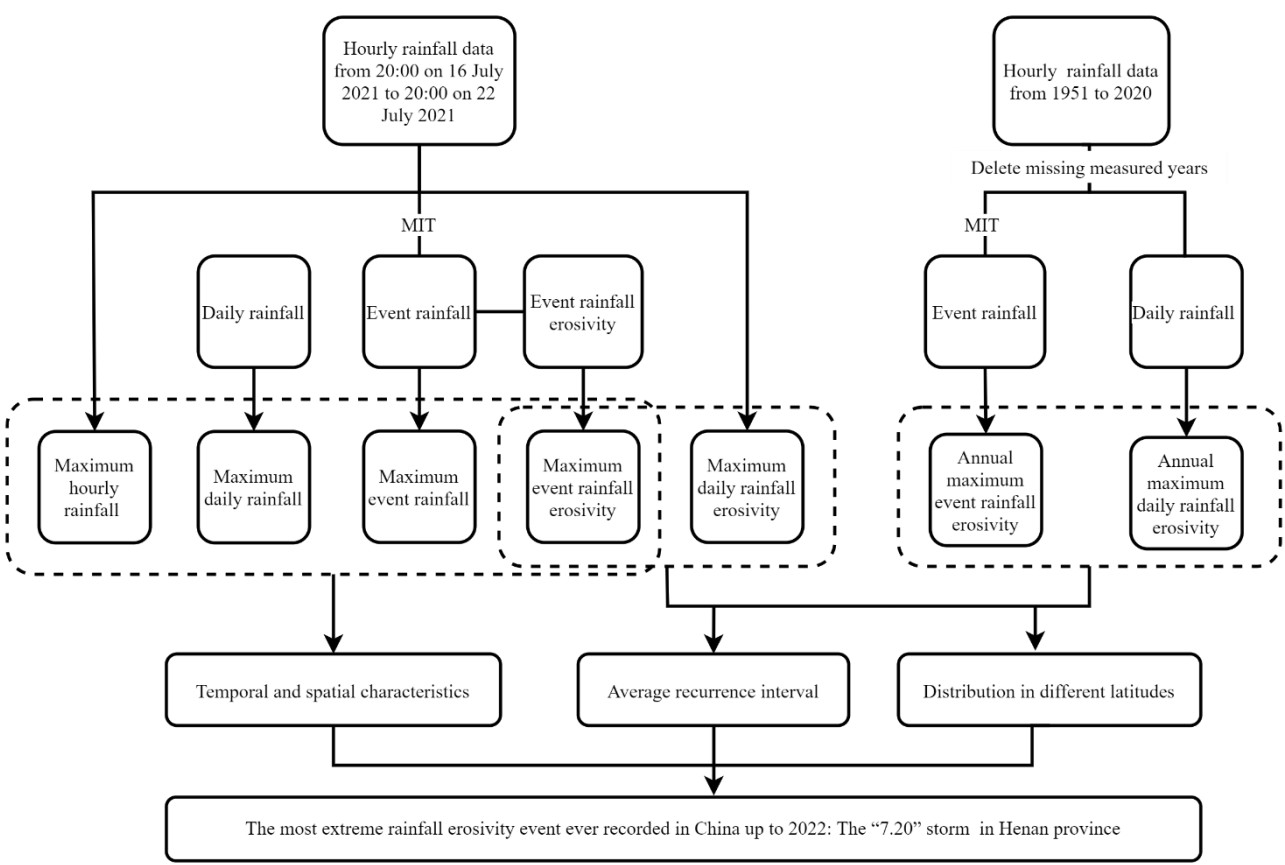

**Figure 2.** Framework for this study

## 3 Results

### 3.1 Temporal and spatial characteristics of the "7.20" storm

*3.1.1 Characteristics of the "7.20" storm*

The extreme event occurred in Henan province between 20:00 on 16 July 2021 and 20:00 on 22 July 2021. The center of the storm is mainly located around Zhengzhou. The storm duration was long and accumulated rainfall was huge. Spatial pattern of accumulated rainfall of the "7.20" storm is shown in Fig. 3a. The top three rainfall stations were Zhengzhou (817.3 mm), Huixian (755.2 mm) and Xinmi (723.5 mm). Additionally, among the 797 automatic meteorological stations in the study area, 58 meteorological stations have accumulated rainfall of more than 250 mm, of which 50 are located in Henan province. Rainfall mass curves for these three stations are shown in Fig. 3b. Obviously, the rainstorm at Zhengzhou meteorological station and Xinmi station contributed more than 50 % of the rainfall in the middle period, while the rainstorm at Huixian station contributed more than 50 % of the rainfall in the last period. Wang et al. (2016) have demonstrated that different rainstorm patterns with rainfall peak in the early, middle and late stages have different effects on soil erosion process, under the natural rainfall conditions. n that study, storms were classified into four patterns: the advanced, intermediate, delayed, and uniform depending when rainfall is most concentrated. The dimensionless durations were separated into three equal periods. Advanced pattern, intermediate pattern and delayed pattern when more than 40 % of the rainfall occurs in the first, second and third periods, respectively. The rainfall temporal distribution is regarded as the uniform pattern otherwise. Wang's research showed that given the same $EI_{30}$, the rainstorm pattern with rainfall peak at the later stage produced more soil loss than the other patterns (Wang et al., 2016).

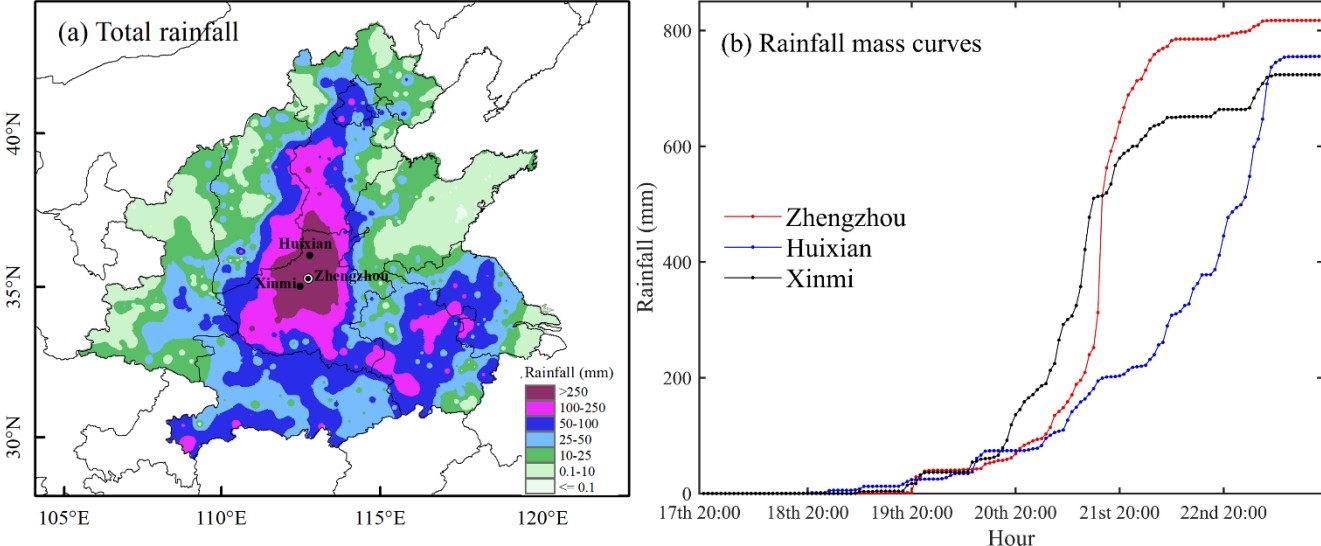

**Figure 3.** A map of total rainfall over the study area, and rainfall mass curves for three stations with the largest rainfall totals.

Spatial pattern of daily rainfall of the "7.20" storm in the study area is shown in Fig. 4. Heavy rainfall mainly occured in the middle and late stages of the event. The maximum daily rainfall (Zhengzhou, 552.5 mm) occurred on 20 July (Fig. 4d), while the storm was most extensive on 21 July (Fig. 4e). The storm is initially concentrated in Anhui province (Fig. 4a), and then dispersed somewhat on 18 July (Fig. 4b). On 19 July, the storm re-appears in the central region of Henan province (Fig. 4c). On 20 July, the storm began to intensify and expand its spatial extent (Fig. 4d). The daily rainfall at 39 meteorological stations exceeded 100 mm, and the daily rainfall of seven meteorological stations exceeded 250 mm on 20 July. On 21 July (Fig. 4e), the center of the storm began to move northward, and the rainfall intensity started to dissipate, and the storm now covered a large area with storm center drifted north to Tangyin (388.2 mm), Henan province, and recorded rainfall at 48 meteorological stations exceeded 100 mm and six meteorological stations exceeded 250 mm. The rainfall decreased considerably by 22 July (Fig. 4f). The storm center was located in the north of Henan province, and the rainfall at 16 meteorological stations exceeded 100 mm.

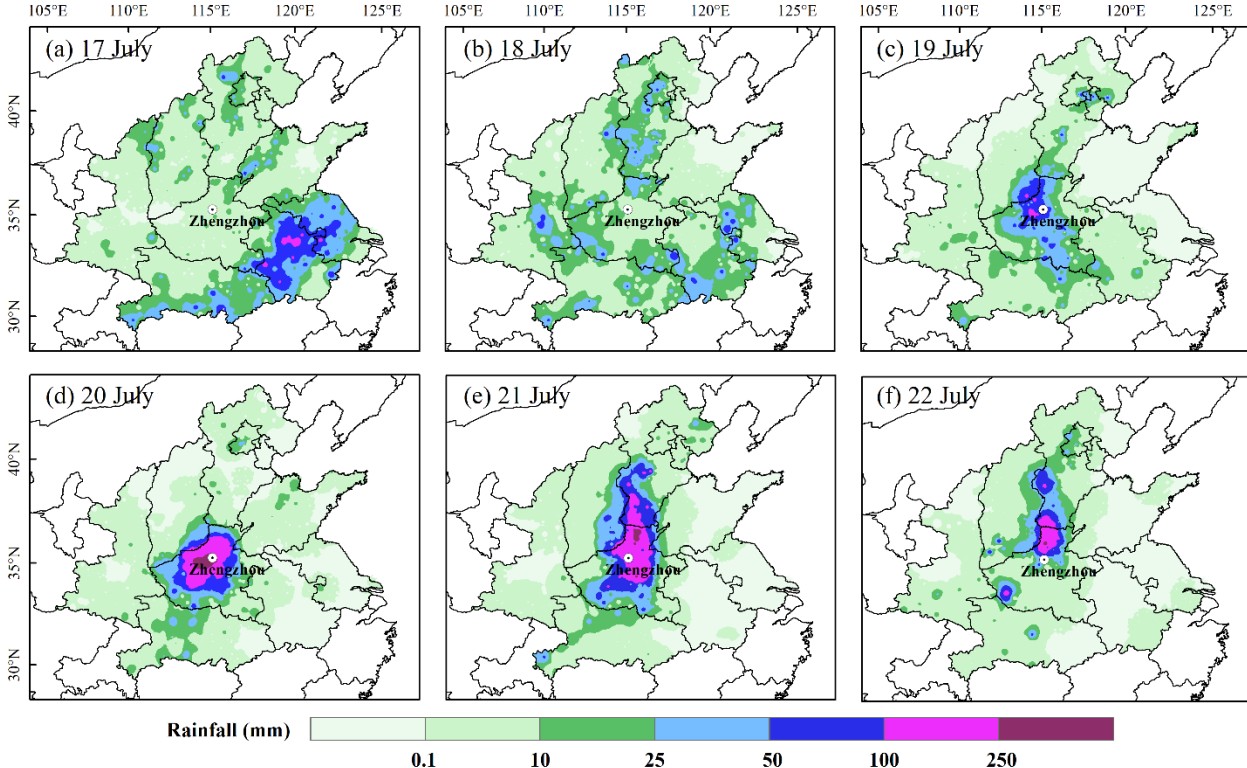

**Figure 4.** Spatial distribution of daily rainfall in the study area. Daily rainfall is rainfall accumulation over 24-hour period, e.g. daily rainfall on 20 July is the total rainfall from 20:00 on 19 July to 20:00 on 20 July.

### 3.1.2 The spatial distribution of rainfall parameters and rainfall erosivity

The spatial distribution of maximum daily and hourly rainfall amount, and maximum event rainfall and rainfall erosivity are shown in Fig. 5. At the center of the storm, a maximum event rainfall amount of 785.1 mm and a maximum daily rainfall amount of 552.5 mm on 20 July were recorded at Zhengzhou meteorological station. From 16:00 to 17:00 on 20 July, maximum hourly rainfall reached 201.9 mm at Zhengzhou meteorological station, and created a new hourly rainfall intensity record (201.9 mm·h$^{-1}$) in mainland China. The maximum event rainfall erosivity in the area with Zhengzhou meteorological station has reached 58,874 MJ·mm·ha$^{-1}$·h$^{-1}$. Due to the uncertainty with the conversion factor, the maximum rainfall erosivity could range from 56,343 MJ·mm·ha$^{-1}$·h$^{-1}$ to 61,405 MJ·mm·ha$^{-1}$·h$^{-1}$.

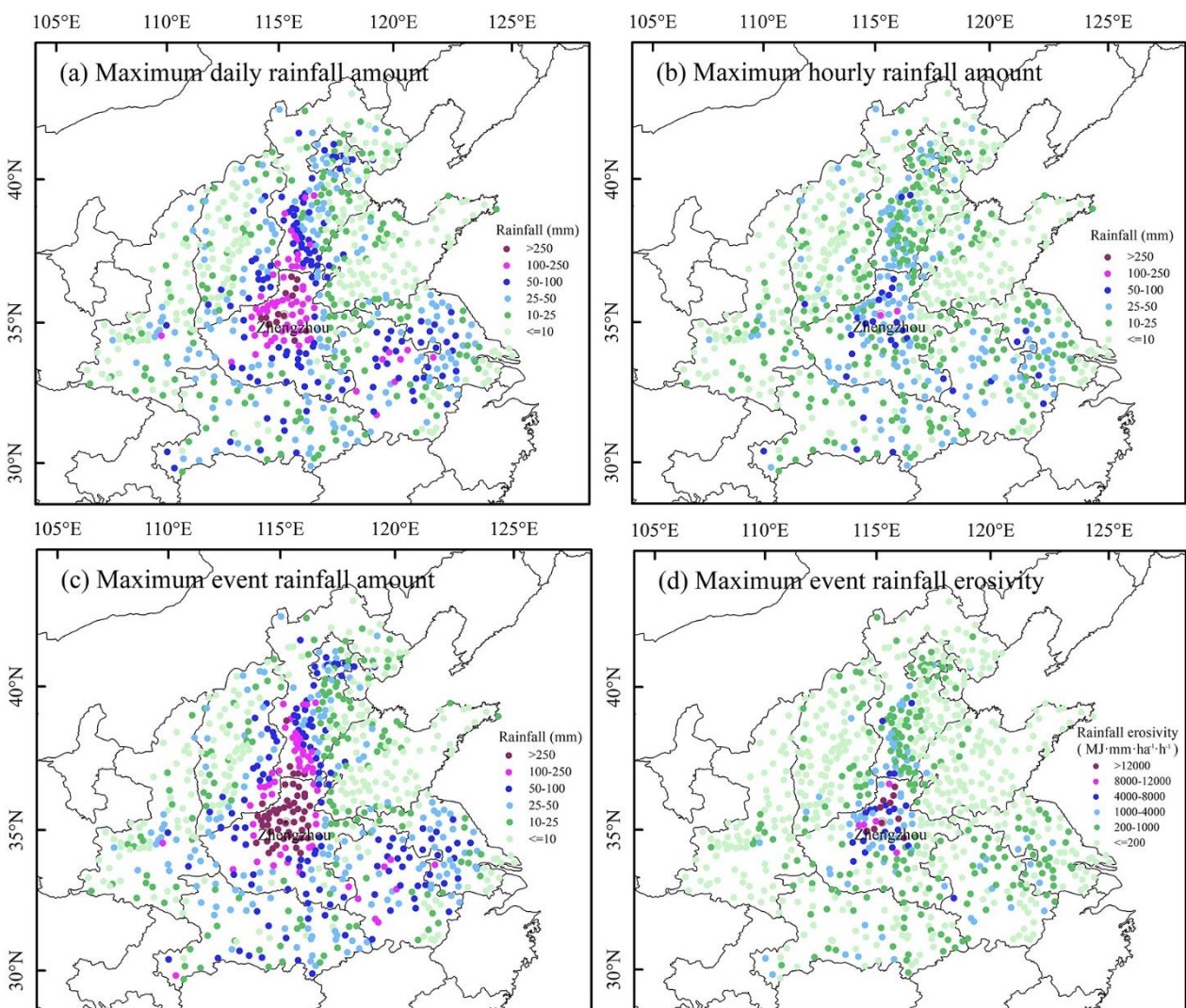

**Figure 5.** Spatial distribution of rainfall amount and rainfall erosivity associated with the "7.20" storm.

*3.1.3 Rainfall's total kinetic energy*

**Table 1.** The composition of average rainfall and energy in different regions from 20:00 on 16 July 2021 to 20:00 on 22 July 2021

| Region | Index | 17th | 18th | 19th | 20th | 21st | 22nd | Total |
|---|---|---|---|---|---|---|---|---|
| Study area (1.33×10⁸ ha) | Mean rainfall (mm) | 12.4 | 10.0 | 6.8 | 8.7 | 11.3 | 5.8 | 55.0 |
| | EN (MJ·ha⁻¹) | 2.6 | 2.0 | 1.3 | 1.7 | 2.3 | 1.1 | 11.0 |
| Henan province (1.66×10⁷ ha) | Average rainfall (mm) | 5.8 | 13.5 | 26.6 | 70.5 | 61.9 | 21.5 | 199.8 |
| | EN (MJ·ha⁻¹) | 2.0 | 3.0 | 4.8 | 13.5 | 15.6 | 7.1 | 46.0 |
| Zhengzhou meteorological station | Average rainfall (mm) | 0.0 | 1.3 | 60.2 | 552.5 | 176.0 | 27.3 | 817.3 |
| | EN (MJ·ha⁻¹) | 0.0 | 0.1 | 12.3 | 144.2 | 40.0 | 4.0 | 200.6 |

The detachment of soil particles from the soil mass and the transportation of detached particles by raindrop impact and surface water flow are two main processes of soil erosion. Rainfall energy reflects the impact of raindrop detachment on the soil. The average rainfall and energy for each meteorological day over different regions of "7.20" storm were listed in Table 1. Comparing the three regions, the average rainfall and EN in the study area on 17 July are higher than those in Henan province and Zhengzhou meteorological station, indicating that the rainstorm center may be outside Henan province at this time. With the movement of rainstorm center, the average rainfall and EN of Henan province and Zhengzhou meteorological station gradually increase. The average rainfall in Henan province reached its peak on 20 July (70.5 mm), but EN reached its peak on 21 July (15.6 MJ·ha⁻¹). The average rainfall and EN of Zhengzhou meteorological station reached the peak on 20 July, which

were 552.5 mm and 144.2 MJ·ha$^{-1}$ respectively. The energy of Zhengzhou meteorological station on 20 July is 11 times of average energy in Henan province.

In summary, an extraordinarily heavy rainfall event occurred in Henan province between 20:00 on 16 July 2021 and 20:00 on 22 July 2021. Among them, the observations of Zhengzhou Meteorological Station show that the maximum event rainfall is 785.1 mm, the maximum daily rainfall is 552.5 mm, the maximum hourly rainfall intensity is 201.9 mm·h$^{-1}$ and the maximum event rainfall erosivity is 58,874 ± 2351 MJ·mm·ha$^{-1}$·h$^{-1}$. The storm is initially concentrated in the southeast of Henan and Anhui provinces, and the rainfall and rainfall intensity reached the peak on 20 July. At the same time, the rainstorm center moved to the north of Henan province with Zhengzhou as the center of the rainstorm. The EN of Zhengzhou Meteorological Station reached 144.2 MJ·ha$^{-1}$ on 20 July. It can be seen that the "7.20" storm has the characteristics of long duration, heavy cumulative rainfall, a wide range of heavy rainfall, and extremely strong short-term rainfall. It is a particularly serious natural disaster that caused serious urban waterlogging, mountain floods, landslides and other disasters, resulting in heavy casualties and serious economic losses.

## 3.2 How extreme is the event recorded at Zhengzhou meteorological station?

### 3.2.1 Frequency of occurrence the maximum daily and event rainfall erosivity

Annual maximum daily rainfall erosivity and the annual maximum event rainfall erosivity in Zhengzhou meteorological station from 1951 to 2020 are shown in Fig. 6 along with fitted LP-III distribution. Using the fitted LP-III distribution, the average recurrence interval of the maximum daily rainfall erosivity of the "7.20" storm is estimated to be about 53,700 years with the lower limit of the 95% confidence intervals is 1229 years, and the ratio of the observed daily erosivity (43,354 ± 1863 MJ·mm·ha$^{-1}$·h$^{-1}$) over 1-in-100-year daily erosivity (6009 MJ·mm·ha$^{-1}$·h$^{-1}$) is 7.21. Similarly, the average recurrence interval of the maximum event rainfall erosivity is estimated to be about 19,200 years with the lower limit of the 95% confidence intervals is 744 years, the observed event erosivity of the "7.20" storm (58,874 ± 2351 MJ·mm·ha$^{-1}$·h$^{-1}$) is 7.75 times larger than the 1-in-100-year event erosivity (7596 MJ·mm·ha$^{-1}$·h$^{-1}$). Based on the 95% confidence interval for the LP-III distribution, the estimated return period of the maximum daily and event rainfall erosivity is most likely to be at least 1229 and 744 years. Evidently, compared with observations in the past decades (1951-2020), the maximum daily and event rainfall erosivity of the "7.20" storm in 2021 is extraordinary, and the event is so rare and extreme that it should be regarded as an outlier among observations in other years.

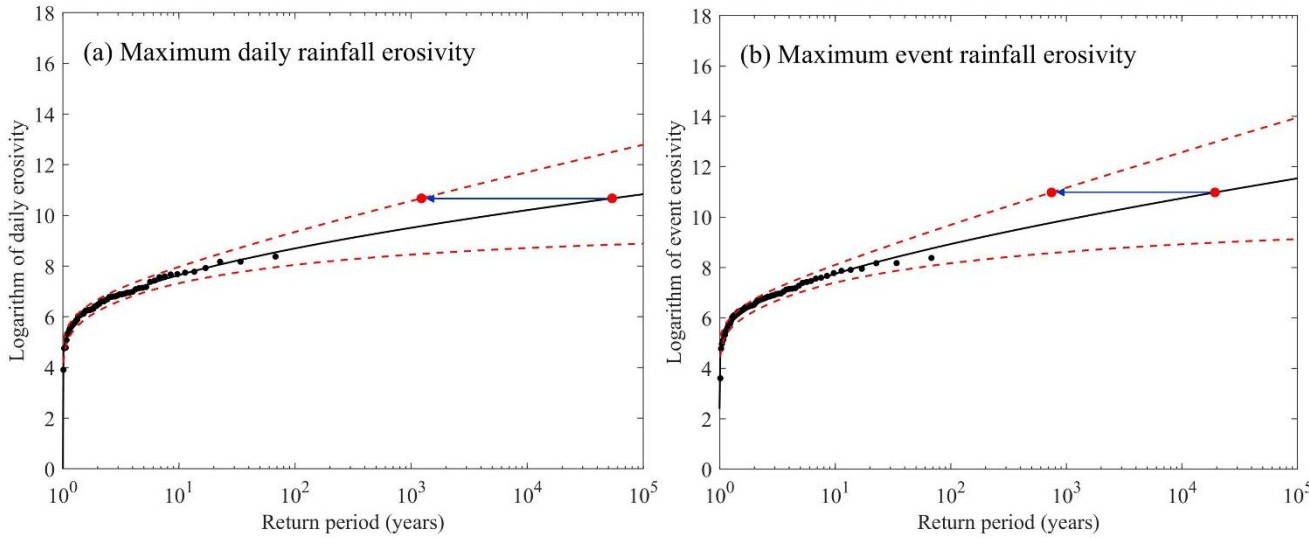

**Figure 6.** The logarithm of observed daily (a) and event (b) rainfall erosivity as a function of the return period assuming LP-III for Zhengzhou meteorological station. Black solid circles are observations from the period 1951-2020, the red solid circles indicate the "7.20" storm in 2021, the red dotted line is the upper and lower limit of 95% confidence interval, and the solid lines in black represent the fitted

P-III distribution using the logarithm of observations from 1951-2020.

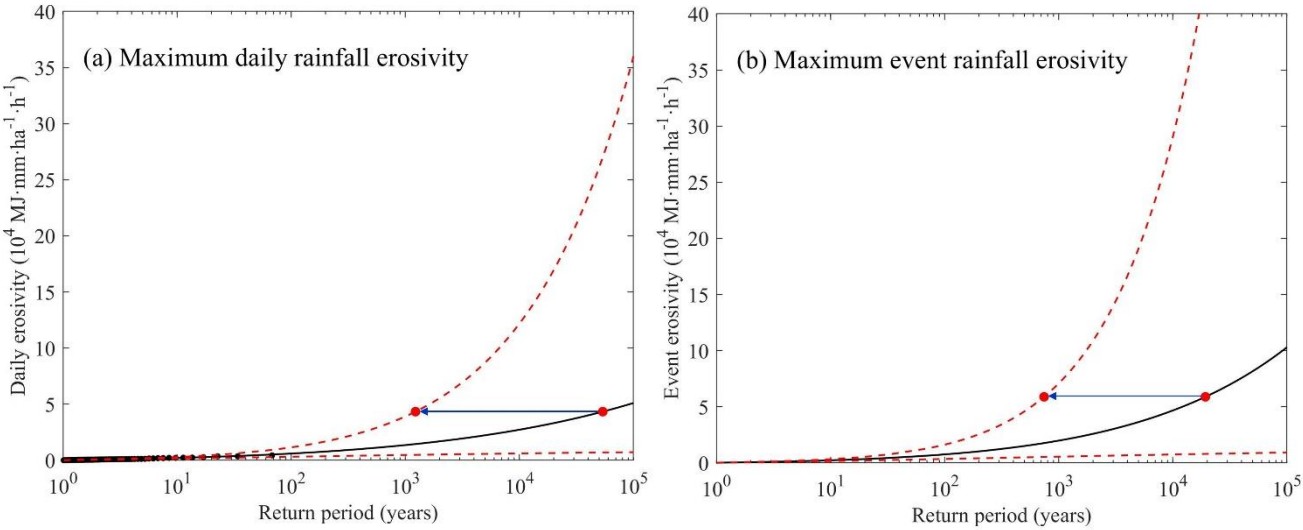

Figure 7. Observed daily (a) and event (b) rainfall erosivity as a function of the return period assuming LP-III for Zhengzhou meteorological station (Performed anti-logarithm conversion for Fig. 6)

The map of the return period of maximum daily and event rainfall erosivity of the "7.20" storm in Henan province is shown in Fig. 8. Similar to Zhengzhou meteorological station, the map of the return period of rainfall erosivity of "7.20" storm in the study area was drawn by fitting the LP-III distribution. The map shows that the return periods of daily (15 stations) and event (17 stations) rainfall erosivity at some meteorological stations exceed 1-in-100-year, mainly in the northern region of Henan province, with Zhengzhou meteorological station as the center of the "7.20" storm.

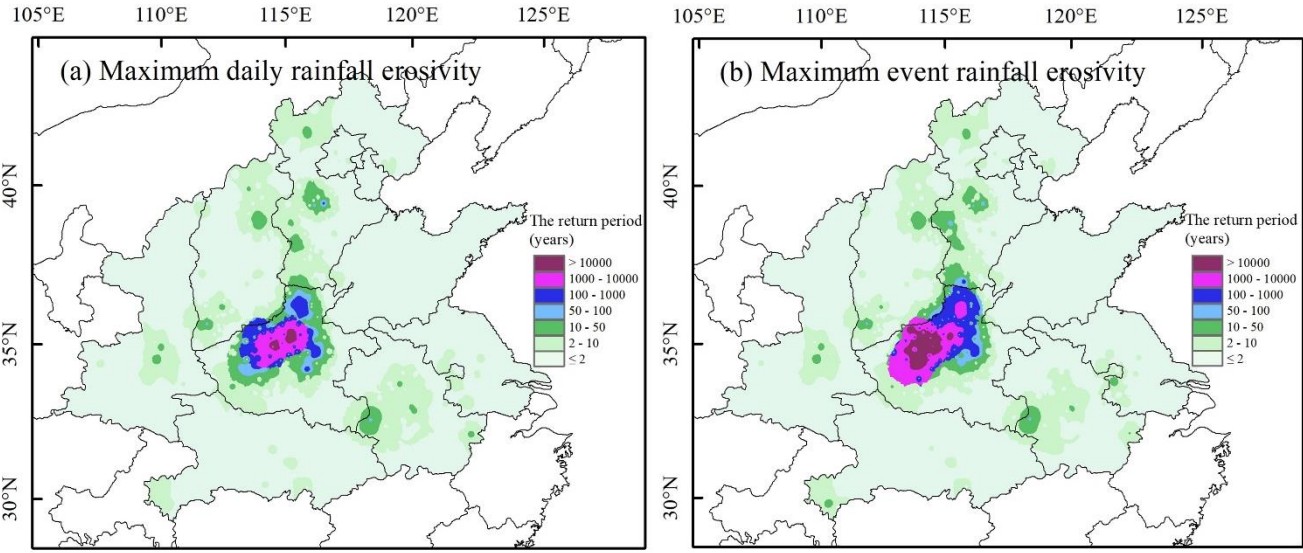

Figure 8. The return period of daily (a) and event (b) rainfall erosivity of the "7.20" storm in the study area.

### 3.2.2 Distribution of the maximum rainfall erosivity in different latitudes

Geographical distribution of the maximum daily rainfall erosivity ever recorded at each of 2420 meteorological stations in China up to 2022 is shown as a function of the latitude in Fig. 9. Envelope curves I and II are drawn for the scatter plot, and the stations and the corresponding daily rainfall and rainfall erosivity values that were used to define these envelope curves are given in Table 2. The two envelope curves overlap at three stations at low latitude and one at high latitude, and the change from curve I to II in the middle latitude is entirely a result of the "7.20" storm in 2021. Prior to the "7.20" storm, curve I shows that the maximum recorded daily rainfall erosivity decreases from about 20°N as the latitude increases, and the maximum

daily erosivity value was 39,345 MJ·mm·ha$^{-1}$·h$^{-1}$, recorded Maoming meteorological station in Guangdong province (21.75°N) on 5 June 2020. Because of the "7.20" storm, the maximum daily rainfall erosivity ever recorded was increased to 43,354 ± 1863 MJ·mm·ha$^{-1}$·h$^{-1}$ or by more than 10 % at Zhengzhou meteorological station (34.72°N) on 20 July 2021.

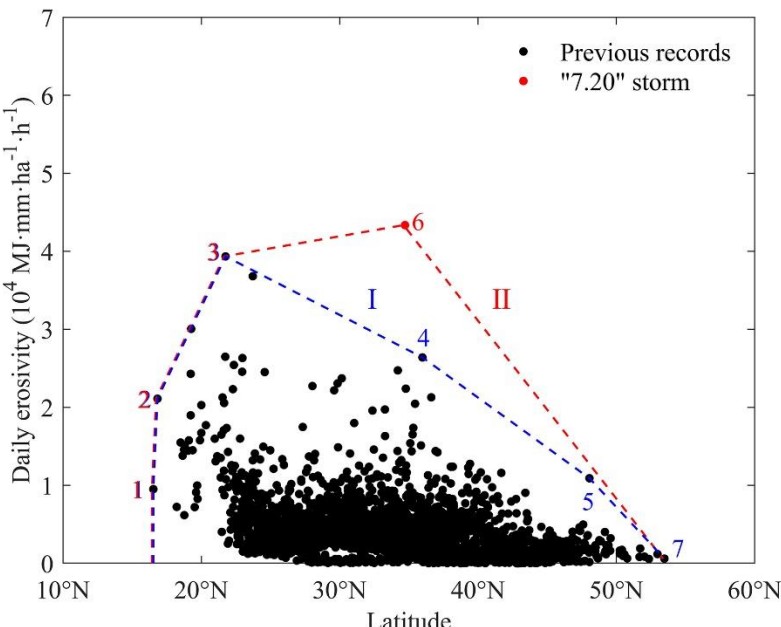

**Figure 9.** The maximum recorded daily rainfall erosivity as a function of latitude for China. The point enclosed by the envelope curve I is the maximum daily rainfall erosivity of each station from 1951 to 2020. The point enclosed by envelope curve II is the maximum daily rainfall erosivity of each station from 1951 to 2021.

**Table 2.** The mean annual rainfall, maximum daily rainfall and rainfall erosivity for stations to define envelope curves.

| ID I | ID II | Station ID | Station name | Latitude | Mean annual rainfall (mm) | Daily rainfall (mm) | Daily rainfall erosivity (MJ·mm·ha$^{-1}$·h$^{-1}$) | Date |
|---|---|---|---|---|---|---|---|---|
| 1 | 1 | 59985 | Shanhu | 16.53 | 1316.0 | 227.6 | 9512 ± 409 | 1980-09-12 |
| 2 | 2 | 59981 | Xisha | 16.83 | 1467.9 | 585.6 | 21,104 ± 907 | 1995-09-05 |
| 3 | 3 | 59659 | Maoming | 21.75 | 1701.7 | 307.3 | 39,345 ± 1691 | 2020-06-05 |
| 4 | | 54848 | Zhucheng | 35.98 | 623.8 | 592 | 26,398 ± 1135 | 1999-08-12 |
| 5 | | 50658 | Keshan | 48.05 | 445.4 | 179.6 | 10,909 ± 469 | 1957-07-15 |
| | 6 | 57083 | Zhengzhou | 34.72 | 566.7 | 552.5 | 43,354 ± 1863 | 2021-07-20 |
| 7 | 7 | 50137 | Beijicun | 53.47 | 385.2 | 77.6 | 603 ± 26 | 2010-07-31 |

Geographical distribution of the maximum event rainfall erosivity ever recorded at each of 2420 meteorological stations in China up to 2022 is shown as a function of the latitude in Fig. 10. Envelope curves I and II are drawn for the scatter plot, and the stations and the corresponding event rainfall and rainfall erosivity values that were used to define these envelope curves are given in Table 3. The two envelope curves overlap at three stations at low latitude and one at high latitude, and the change from curve I to II in the middle latitude is entirely a result of the extreme "7.20" storm in 2021. Prior to the "7.20" storm, curve I shows that the maximum recorded event rainfall erosivity decreases from about 20°N as the latitude increases, and the maximum ever event erosivity value was 41,537 MJ·mm·ha$^{-1}$·h$^{-1}$, recorded Maoming meteorological station in Guangdong province (21.75°N) from 20:00 on 20 May 1987 to 18:00 on 22 May 1987. Because of the "7.20" storm, the maximum event rainfall erosivity ever recorded was increased to 58,874 ± 2351 MJ·mm·ha$^{-1}$·h$^{-1}$, or an increase of more than 40 % at Zhengzhou meteorological station (34.72°N) on 20 July 2021.

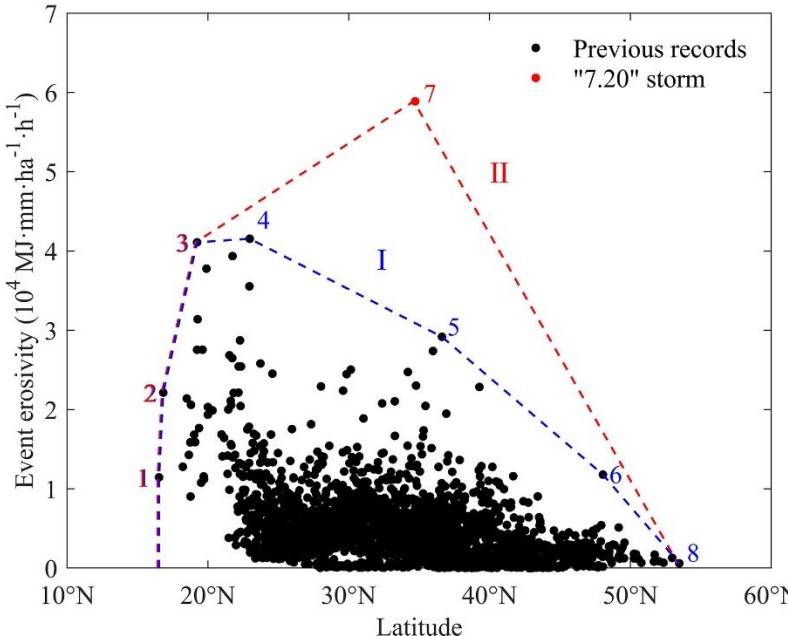

**Figure 10.** The maximum recorded event rainfall erosivity as a function of latitude for China. The point enclosed by the envelope curve I is the maximum event rainfall erosivity of each station from 1951 to 2020. The point enclosed by envelope curve II is the maximum event rainfall erosivity of each station from 1951 to 2021.

**Table 3.** The mean annual rainfall, maximum event rainfall and rainfall erosivity for stations to define envelope curves.

| ID | | Station ID | Station name | Latitude | Start date and time | End date and time | Mean annual rainfall (mm) | Event rainfall (mm) | Event rainfall erosivity (MJ·mm·ha⁻¹·h⁻¹) |
|---|---|---|---|---|---|---|---|---|---|
| I | II | | | | | | | | |
| 1 | 1 | 59985 | Shanhu | 16.53 | 1980-09-11 11:00 | 1980-09-12 8:00 | 1316.0 | 288.2 | $11{,}446 \pm 492$ |
| 2 | 2 | 59981 | Xisha | 16.83 | 1995-09-05 8:00 | 1995-09-06 23:00 | 1467.9 | 625.5 | $22{,}135 \pm 951$ |
| 3 | 3 | 59855 | Qionghai | 19.23 | 2010-10-01 22:00 | 2010-10-08 15:00 | 2021.7 | 1433.3 | $41{,}083 \pm 1766$ |
| 4 | | 59500 | Haifeng | 22.97 | 1987-05-20 20:00 | 1987-05-22 18:00 | 2407.5 | 987.3 | $41{,}537 \pm 1785$ |
| 5 | | 53892 | Handan | 36.62 | 1963-08-03 3:00 | 1963-08-06 1:00 | 478.8 | 748.1 | $29{,}174 \pm 1254$ |
| 6 | | 50658 | Keshan | 48.05 | 1957-07-15 14:00 | 1957-07-15 24:00 | 445.4 | 199.5 | $11{,}794 \pm 507$ |
| | 7 | 57083 | Zheng-zhou | 34.72 | 2021-07-18 8:00 | 2021-07-21 10:00 | 566.7 | 785.1 | $58{,}874 \pm 2351$ |
| 8 | 8 | 50137 | Beijicun | 53.47 | 2010-07-30 23:00 | 2010-07-31 14:00 | 385.2 | 77.6 | $603 \pm 26$ |

A large number of studies have shown that the mean annual rainfall and rainfall erosivity, i.e. the R-factor, decrease from southeast to northwest in China (Yin et al., 2019; Yue et al., 2022), that is, the mean annual rainfall and rainfall erosivity is the highest at low latitude in China. Like rainfall, the average rainfall intensity for given storm duration also tends to be high at low latitude, and low at high latitude in China (Kong et al., 2017). Thus, one would expect that maximum daily and event rainfall erosivity tends to decrease with the latitude, a trend largely supported by the envelope curve I in Fig. 9 & 10. The "7.20" storm may have fundamentally changed the nature and distribution of extreme daily and event erosivity in China as we knew them up to now. This is consistent with the research of Wang and Luo (2006), and the storm extreme value does not always conform to the pattern of decreasing from low latitude to high latitude. For example, based on measured and surveyed rainfall records, the maximum 24-hour rainfall depth occurred at Linzhuang in Henan province in the mid-latitude on 5–7 August 1975 (Ding., 2015). Occurrence of this "7.20" storm in 2021 around Zhengzhou has important implications. First,

Figure. 9 & 10 suggest that extreme event erosivity may be the highest in mid-latitude around 35°N despite the fact the mean annual rainfall and rainfall intensity are by no means the highest in mid-latitude in China. Second, the "7.20" storm was out of ordinary that the event was seemingly unrelated to the underlying climatology. Finally, the "7.20" storm has led us to realize that such extreme erosive events could and may occur anywhere in eastern China with further implications for soil conservation planning.

## 4 Discussion

The above analysis shows that the "7.20" storm is the largest in terms of the rainfall erosivity among 2420 meteorological stations in mainland China up to 2022. However, there are limitations and uncertainties in our assessment due to KE-I equations, $EI_{30}$ conversion factors, and probability distributions used.

Firstly, soil erosion processes are related to rainfall kinetic energy, which is a function of the size and fall-velocity of raindrops. Different KE-I relationships were recommended in different version of the USLE, and yet more location-specific KE-I relationships were noted for various regions around the world (van Dijk et al., 2002). Using different KE-I relationships, including those for the USLE, RUSLE, and from van Dijk et al. (2002), in addition to the RUSLE2 equation adopted for the study shows that other KE-I relationships would underestimate kinetic energy. Storm energy for the "7.20" storm using other KE-I relationships was 3.1% to 8.2% smaller than reported in the study, and the annual maximum event kinetic energy from 1951 to 2020 would differ by -16.9% to 28.7% from that reported in the study (Table S1.2). The uncertainty associated with different KE-I relationship does not increase with the magnitude of the rainfall event as shown in Fig. S1.1. Similarly, there are considerable differences in the estimated return periods of the event in terms of rainfall erosivity using different KE-I equations (Fig. S1.2). The return period of the "7.20" storm varied from about 20 thousand to more than 50 thousand years. The relatively small difference in event KE can lead to considerable differences in the return period for such an extreme event when the KE value of event exceeded all other KE values for the site by at least an order of magnitude. These large uncertainties associated with the return period of extreme precipitation have been noted in Germany (Grieser et al., 2007).

Secondly, rainfall erosivity is usually calculated using long term precipitation records from rain gauges, and depends strongly on the temporal resolution of the precipitation data used. Data at higher temporal resolution would be higher desirable to compute rainfall erosivity is high temporal resolution. However, such data are in short supply, short in length and sparse in spatial coverage. R-factor values decrease with decreasing temporal resolution because intensities are reduced when precipitation amount is aggregated over longer time intervals (Fischer et al., 2018). Therefore, it is necessary to use conversion factors to adjust the computed $EI_{30}$ value using data of low temporal resolution. The conversion factor for the 1-in-10-year $EI_{30}$ computed with the one-min resolution rainfall data is 1.489, which was appropriate for evaluating extreme rainfall erosivity in this study. To allay the reviewer's concerned, we collected one-min temporal resolution rainfall data from Zhengzhou Meteorological Station from 2005 to 2016, The annual maximum $EI_{30}$ values estimated using one-min and one-hour data were compared (Fig. S2.2). The conversion factor for the annual maximum $EI_{30}$ at Zhengzhou meteorological station is 1.974, which is very close to the conversion factor of 1-in-10-year $EI_{30}$ is 2.029.

Finally, the estimated return period depends on the selected probability distribution function. Different probability distribution functions can produce quite different estimates for large return periods (Laio et al., 2011). Three frequency distributions were considered and tested, including Generalized extreme value (GEV), P-III, and LP-III was found to be the most appropriate (Table S3.1). All the three distributions fitted the observations well, and performance indicator values do not suggest a single distribution that consistently and significantly superior to others (Table S3.2). The return period estimated by the three probability distributions are quite different. The average recurrence intervals of the maximum event rainfall erosivity of GEV and P-III for "7.20" storm exceeds 340,600 years, which is far greater than reported in the study. The estimated return

period of around 20,000 years for the "7.20" storm is conservative. The estimated return period would be much higher if we use other KE-I equations and other probability distributions. Given LP-III was widely recommended for extreme precipitation and flood events in China (Chen et al., 2012), LP-III was used to assess the return period of this "7.20" storm. Estimating return periods comes with large uncertainties, especially for return periods exceeding the length of the observational record (Bloemendaal et al., 2020).

## 5 Conclusions

This study assessed an extreme rainfall event in Henan province from 20:00 on 16 July 2021 to 20:00 on 22 July 2021, using hourly rainfall data from 796 stations in Henan and surrounding provinces. Based on hourly rainfall data of 2420 meteorological stations in China from 1951 to 2021, the annual maximum daily and event rainfall erosivity of Zhengzhou meteorological station were fitted with the LP-III distribution to evaluate magnitude and frequency of occurrence of this extreme event in terms of rainfall amount and erosivity values. The following conclusions can be drawn as a result of this research:

(1) The maximum event rainfall (785.1 mm), maximum daily rainfall (552.5 mm), maximum hourly rainfall intensity (201.9 mm·h$^{-1}$) and maximum event rainfall erosivity ($58,874 \pm 2351$ MJ·mm·ha$^{-1}$·h$^{-1}$) of "7·20" storm all occurred and were recorded at Zhengzhou meteorological station. The period of the highest rainfall intensity was mainly concentrated in the middle and late stages of the storm, reaching its peak on 20 July, producing a daily total of 144.2 MJ·ha$^{-1}$ energy.

(2) Based on long-term observations for the period 1951-2020 and the fitted LP-III distribution, the "7.20" storm was estimated to have an average recurrence interval in excess of 10,000 years, and the annual maximum daily and event rainfall erosivity were about seven times larger than 1-in-100-year erosivity values.

(3) This extreme event recorded at the Zhengzhou meteorological station has set a new record for daily and event rainfall erosivity values in mainland China. The "7.20" storm in 2021 was extremely rare, and suggested to us that extreme erosive events could and may occur anywhere in eastern China, rather than in low latitude with high rainfall amount and rainfall intensity as we previously knew and expected.

*Data Availability*. Observed hourly rainfall data from National Meteorological Information Center of the China Meteorological Administration (NMIC, http://data.cma.cn).

*Author contributions*. Y Xiao, S Yin and B Yu conceived and designed this study. Y Xiao undertook data analysis and interpretation, prepared all the diagrams, and drafted the manuscript. S Yin and B Yu guided data analysis and interpretation. All authors reviewed and edited the manuscript.

*Competing interests*. The authors declare that they have no conflict of interest.

*Disclaimer*. Publisher's note: Copernicus Publications remains neutral with regard to jurisdictional claims in published maps and institutional affiliations.

*Financial support*. This research has been supported by the National Key Research and Development Program of China (grant no. 2021YFE0113800), National Natural Science Foundation of China (grant no. 41877068) and the project "ACRP 13: Soil erosion in Austria -from mean to extreme (EROS-A)" funded by the Klima- und Energiefonds.

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
