# Peer review of "The most extreme rainfall erosivity event ever recorded in China up to 2022: The “7.20” storm in Henan province"

_Hydrology and Earth System Sciences, 2022_

## Author Response (AR1)

**Author's response to Editor's decision and comments from Reviewers**

**The most extreme rainfall erosivity event ever recorded in China:**
**The "7.20" storm in Henan province**

Yuanyuan Xiao, Shuiqing Yin, Bofu Yu, Conghui Fan, Wenting Wang, Yun Xie

We would like to thank the Editor and Reviewers for all their comments and suggestions. In the revised version, we improved the text and figures, addressing all the issues raised by the reviewers. We hope this revision is satisfactory for the further processing of this paper.

**Dear Editor**,

We would like to thank you very much for your feedback. We revised the manuscript in detail, aiming to answer all reviewers' points. We would like to submit the revised version of our paper "The most extreme rainfall erosivity event ever recorded in China: The "7.20" storm in Henan province" by Yuanyuan Xiao, Shuiqing Yin, Bofu Yu, Conghui Fan, Wenting Wang and Yun Xie.

We believe that we followed all the proposed corrections by the reviewers that substantially improved the article and we are looking forward to hearing from you with respect to the review process.

We provide answers to comments from Reviewers below. For clarity, each answer is structured as follows: (1) RC# comments from Referees (black), (2) Author's response (blue).

Thank you for your time and consideration.

Sincerely,

Yuanyuan Xiao

**RC1: 'Comment on hess-2022-351', Anonymous Referee #1, REPLY**

**General comments**

In the submitted paper authors investigate the characteristics of one extreme rainfall event (20th July, last year) in comparison to spatial and temporal rainfall erosivity characteristics in China. The paper is interesting and within the scope of the HESS journal. However, there are several points that could be improved.

Author response: Thank you for your careful review and valuable suggestions. Here we present our responses in blue.

Firstly, the reported rainfall amounts are relatively extreme. Hence, more details about the measuring equipment used to measure rainfall (and accuracy of these instruments) should be reported since this could have an effect on the measured rainfall amounts (can at least for this extreme event uncertainty be estimated).

Author response: We agree that the accuracy of the instruments is important. Observed hourly rainfall data from 1951 to 2021 for 2420 meteorological stations in China were collected with siphon rain gauges (before about 2000~2005 varying across stations) and tipping bucket rain gauges since. The data had been quality-controlled and the homogeneity of the data had been checked by National Meteorological Information of Center of China Meteorological Administration. The details of the data will be added in the paper.

Secondly, the reported results are sensitive to the selected empirical equation used to calculate the energy. Hence, are there any other data available (e.g., optical disdrometer) measurements that could be used to validate these calculations in order to make less uncertain rainfall erosivity estimates?

Author response: As there was no direct measurements of kinetic energy, kinetic energy as a function of rainfall intensity, known as KE-I equation was used to estimate storm energy as recommended for RUSLE2, which is widely used as an empirical model.

Thirdly, the reported frequency analysis is missing uncertainty estimation (confidence intervals).

Author response: Uncertainty analysis is important for such a rare rainfall event. We carefully compare the simulation effects and confidence intervals of several extreme value distributions including Pearson-III, Log-Pearson-III, GEV and Gumbel, and we found Log-Pearson-III generated the most reasonable estimate. The logarithm of the observations was first taken before Pearson-III was applied, and an anti-logarithm transformation was performed thereafter. Based on Log-Pearson-III, the estimated return period of the "7.20" storm in terms of its rainfall erosivity is about 10,072 years. Based on the 95% confidence interval for the log-Pearson-III distribution, the estimated return period is most likely to be at least 516 years (Figure 1). Therefore, Log-Pearson-III will be used to replace GEV and its uncertainty associated with the estimated return period will be added in the revised version.

[Figure]

Figure 1. The logarithm of observed daily (a) and event (b) rainfall erosivity as a function of the empirically fitted return period for Zhengzhou meteorological station (black solid circles are observations from the period 1951-2020, the red solid circles indicate the "7.20" storm in 2021, the red dotted line is the upper and lower limit of 95% confidence interval, and the solid lines in black represent the fitted P-III distribution using the logarithm of observations from 1951-2020)

Fourthly, I am not sure what is the purpose of envelope curves (see also specific comment below). I would suggest to add some specific details about the impact of this extreme event on soil erosion (some measurements perhaps, if available) or at least on the sediment concentrations in rivers (some measurements) or something similar. Hence, could you say that extreme erosive events (with return period over 100,000 years) also leads to soil erosion rates with similar recurrence interval (the same for sediment transport rates).

Author response: Envelope curves were drawn to show the maximum event rainfall erosivity at different latitudes. In previous studies, the average rainfall and rainfall erosivity in China decreased from southeast to northwest. In the south of China, the frequency of rainstorm is higher than that in the north, and the likelihood of experiencing extreme rainfall erosivity in the south is also the highest, but this is no longer the case based on our research. The "7.20" storm indicates that extreme rainfall erosivity could also occur in mid-latitude in China. That is why these envelope curves were drawn. Since soil erosion and sediment concentrations in rivers are determined not only by rainfall but also by topography, soil erodibility, vegetation cover and so on, we don't expect a similar recurrence interval for the amount of soil erosion. However, we believe that this is a good suggestion to add some specific details about the impact of extreme rainstorm on soil erosion. Unfortunately, due to difficulties in obtaining data, we are unable to prove this conjecture.

Finally, some specific comments are provided below.

**Specific comments**

Figure 1: Maybe you could more clearly indicate Henan province in this figure.

Author response: We have marked the location of provinces in Figure 1.

Equation 1: You should cite the original source of this equation. Additionally, what is the sensitivity of results with respect to the selection of equations (1) and (4).

Author response: We have provided the original references of the two equations in the paper. The sensitivity of the empirical equation selected for calculating rainfall erosivity has been explained in the General comments.

Lines 132-135: Please provide more details about interpolation method used

Author response: We used the Inverse Distance Weighted (IDW) to interpolate the point data to the region. We will explain this in detail in the revision.

Equation 5: Please provide the original reference.

Author response: We have provided the original reference of equation 5 in the paper.

Line 147: Shape, scale and location parameters and not position parameter.

Author response: We have changed 'position,' to 'location' in line 147.

Equations (6)-(7): Please double check it, I am not sure if these are correctly written.

Author response: We have decided to change the method of frequency analysis. We will carefully check the accuracy of the formula.

Equations (8)-(17): I am not sure if these need to be reported in a paper about rainfall erosivity. More details about the rainfall erosivity calculation procedure and measurements could be provided instead.

Author response: After the paper is revised, if necessary, we can include this section as supplementary material.

Line 186: "It showed". Is this referring to Want et al. (2016) study?

Author response: Yes, "It showed" refers to the research of Wang et al. (2016) study. We will revise it to make readers understand it better.

Figure 3: It is not clear how was Figure 3a created, is this station-based data interpolated or this is from other source (radar)?

Author response: Figure 3 is obtained by IDW interpolation based on observation data, which will be explained in detail in the paper.

Figure 4: The same as for Figure 3.

Author response: Figure 4 is obtained by IDW interpolation based on observation data, which will be explained in detail in the paper.

Figure 5d: Here these results are probably quite sensitive to the selection of the empirical equations used to calculate the rainfall erosivity. It would be nice to elaborate a bit about this issue.

Author response: This problem has been explained in the general comments. More explanation will be added in the revised version.

Table 1: Why ha? I suggest to use either km2 or 1000*km2 or something similar. Also in this table you are comparing areal rainfall erosivity with station-based (gauge, probably 200 cm2 or something similar).

Author response: Yes, we are comparing areal kinetic energy with station-based. In section 2.2.2, the unit of kinetic energy is MJ·ha$^{-1}$, the standard unit of measurement used for the USLE/RUSLE, while the unit of rainfall erosivity is MJ·mm·ha$^{-1}$·h$^{-1}$, so we also use ha for the area unit in Table 1. The unit of 'ha' has been widely and conventionally used in agriculture.

Figure 6: Here you clearly need the conference intervals. I am not sure if you could just say that the return period of this event is exactly 154,154 years. Additionally. You should note that in (flood) frequency analysis there are usually some specific rules about the longest return period that could be estimated based on specific data length (sample size). Different rules can be found in the literature. At least some discussion about this should be added.

Author response: Obviously, we cannot say that the return period of this event is exactly 154,154 years. It is quite uncertain to use about 70 years of data to estimate the return period in excess of 100,000 years. We have used the 95% confidence interval to assess the uncertainty.

Section 3.2.2: I am sorry but I do not completely understand the purpose of defining these envelope curves? How could these be used? It is clear that the shape of the "curve" is defined by the extreme events (as authors also indicate in the last sentences of this paragraph) that are a result of stochastic process.

Author response: Please refer to the reply in the general comment.

**RC2: 'Comment on hess-2022-351', Anonymous Referee #2, REPLY**

**General comments**

The authors studied and characterize a record extreme rainfall observed in China, in July 2020 in terms of its erosivity. The authors did a great job regarding the conciseness of the paper, and it is within the scope of the HESS journal. I have some concerns that need to be addressed before the paper is considered for publication, mainly regarding the frequency analysis.

**Specific comments:**

There are many instances of grammatical and incomplete sentences (some examples are given below). I, therefore, recommend a full language review.

The authors used GEV for the frequency analysis, but a lot of information is missing. For instance, the estimated parameters (most importantly the shape parameter) are presented and the uncertainty i.e confidence intervals are missing in the plots (Figure 6 ). The plot is also so condensed below the 100-year level to make any comments regarding the quality of the fit. The return period of the largest value is given as a point value, at least the lower and upper bounds should be given knowing that a lot of uncertainty is expected given that only around 70 years of data is used to infer a 100,000-year return period.

Author response: Thank you for your careful review and valuable suggestions. According to your suggestions, we carefully compare the simulation effects and confidence intervals of several extreme value distributions including Pearson-III, Log-Pearson-III, GEV and Gumbel, and we found that Log-Pearson-III generated the most reasonable estimate. Therefore, Log-Pearson-III is used to replace GEV and its uncertainty estimate will be added in the revised version.

**Technical corrections**

L 101-102 : "Hourly rainfall data from 1951 to 2020 were as historical data" > the sentence is not complete

Author response: The sentence is revised to read 'Hourly rainfall data from 1951 to 2020 were used to calculate the annual maximum daily and event rainfall erosivity.'

The caption of figure 2 seems too short

Author response: Figure 2 caption is changed to "Workflow of this study".

Figure 6: The caption : "Observed daily (a) and event (b) rainfall erosivity as a function of the empirical and return period …." > this seems incomplete. Empirical what??

Author response: The caption of Figure is revised to read 'Figure 6. The logarithm of observed daily (a) and event (b) rainfall erosivity as a function of the empirically fitted return period for Zhengzhou meteorological station'.

The relevance of Figures 7 and 8 should be made more clear, also different colors could be used to distinguish the two curves

Author response: We have used two different colors to distinguish the envelope curves.

L 265: "Post the "7.20" rainstorm, the ….. " > This is not clear

Author response: We have changed 'Post the "7.20" rainstorm,' to 'Because of the "7.20" storm,' in Line 265.

**RC3: 'Comment on hess-2022-351', Anonymous Referee #3, REPLY**

**General comments**

This paper aims to characterize the heaviest recorded rainfall in Henan, China, in recent times. Towards this, the available rainfall data is well presented and compared with historical data of the past 70 years from meteorological stations all over Mainland China. However, in terms of stated objectives, specifically 'characterizing spatial and temporal distribution', the paper seems to fall short.

Author response: Thank you for your careful review and valuable suggestion. We included the characteristics of the storm and its temporal and spatial distribution as background information. This is not one of the research objectives, and we have removed this from the list of objectives in the revised manuscript.

In the frequency of occurrence analysis (section 3.2.1), the GEV distribution is fitted with '7.20' rainstorm. This being so rare (which is stated as an outlier in line 235) could be biasing the curve as far as the previous occurrences are concerned. To complement this analysis, it would be relevant to assess GEV parameters without accounting the '7.20' event and quantify the difference when the extreme is considered. Further, the uncertainty associated with GEV parameters could also be illustrated.

Author response: We agree with you on this. In the paper, we used observed largest events each year other than the "7.20" storm for curve fitting (e.g. black solid circles in Figure 6). Based on the fitted curves, we can estimate the return period (e.g., red solid circles in Figure 6) of the "7.20" storm. The uncertainty associated with this estimate will be added in the revised version.

Since the frequency analysis is only performed at Zhengzhou meteorological station, the result cannot be completely said to be characterizing spatial distribution. Maybe, it will be worth exploring similar

behaviour at other stations available. A map indicating the return period of the '7.20' storm for remaining stations at Henan province could be helpful in this regard as well.

Author response: We agree that it would be useful to show the extreme nature the "7.20" storm using data from Zhengzhou Meteorological Station alone. We will include a map of the return period of the "7.20" storm for all stations in Henan Province.

Here are some of the other concerns and suggestions for improving the paper. Though the paper is relevant in the application of existing techniques to an extreme event, it could benefit from clear definition of purpose as well as innovativeness in analysis. Addressing these, acceptance of the paper is recommended after a major revision

**Detailed comments**

**(1)Introduction**

- The importance of establishing rainfall erosivity values is rightfully stated. However, it is also dependent on how the rainfall kinetic energy is obtained. Since the paper is not using directly measured energy but rather a derived value from rainfall rate (eq.1, RUSLE2), I would suggest mentioning probable biases expected in this approach. It would be worth stating that drop size distribution (dsd), as well as drop velocity, are following certain assumptions in this approach rather than being directly measured.

Author response: We agree that there is uncertainty in the estimated kinetic energy from rainfall intensity. We have explained this in the revised version.

- line 75: 'The strengthen and '– was this supposed to be 'strengthened'

Author response: We have changed 'strengthen' to 'strengthened' in line 75.

**(2.1) Data source and pre-processing**

- If the historical data is available to the general public, I would suggest putting a link to that information. More information on nature of data is also desired here

Author response: We have added more detailed information on the nature of precipitation data. The link to access the data will be added.

- Similar comment on rest of the data; please insert information on devices used, quality of data as well as resolution available

Author response: We have added data sources, observation equipment, quality control measures and other information in the revised version.

- Though later mentioned in section 2.2.3, I would suggest including the number of years discarded here as well, as it is defined at this section. If it could be clarified that missing years didn't have extreme events (to the level that it could bias current analysis) that would be good too.

Author response: We agree that the treatment of missing years is very important for the extreme value analysis. We have checked this using daily precipitation observation from rain gauges and provide clear explanations in the revised version.

- line 100: 'Hourly rainfall data from 1951 to 2020 ….' Please reframe this sentence.

Author response: The sentence has been revised to read 'Hourly rainfall data from 1951 to 2020 were used to calculate the annual maximum daily and event rainfall erosivity.'

- Figure 1: seems like the scale should be outside the inset area since it corresponds to the mainland than nine-dash line map.

Author response: We have revised the scale in Figure 1.

**(2.2) Framework of the study**

- There are some minor continuation issues in the usage of the terms 'maximum event rainfall amount' and 'maximum daily rainfall amount'. For example, in line 115 they are 'maximum event rainfall' and 'maximum daily rainfall' while in figure 2 they are 'maximum event amount' and 'maximum daily amount'. The former is again used in figure 5. Though minor, it's better to follow single usage for readability.

Author response: The terms "maximum event rainfall" and "maximum daily rainfall" follow a single usage. We have revised the terms throughout the manuscript as follows: "maximum event rainfall" and "maximum daily rainfall".

- Figure 2: misplaced 'and' in the top row 'Hourly and rainfall data..'

Author response: We have deleted "and" in "hourly and rainfall data" in Figure 2.

- line 125: er unit is repeated (make sure to keep the format used throughout the paper while removing the extra entry of unit)

Author response: We have maked sure to keep the same format used throughout the paper while removing the extra entry of unit.

- line 130: 'Henan province' seems to be repeated in the text as it is already covered in the study area

Author response: Our study area covers Henan Province, but considering that the "7.20" storm mainly occurs in Henan Province, we have compared the rainfall and kinetic energy of the whole study area with Henan Province.

- line 150: eq.8 is using 'E' for expected values. As 'E' is already used as total energy, another variable would be better

Author response: We have revised the manuscript accordingly.

- line 160: This is a follow-up comment to the one mentioned in the introduction part. The usage of various moment estimators is known to have an effect according to the type of dsd (and hence to energy and erosion), thanks to the measuring instrument. If it is relevant in this data used, think about including a line at the introduction or here.

Author response: We have explained this in the revised version.

- line 165: eq.17 is placed after its description

Author response: We have adjusted the position of equation 17 and its description in line 165.

**(3.1) Temporal and spatial characterization of the "7.20" rainstorm**

- Since "7.20" storm is defined at the start of this chapter, it would be better to follow that terminology for the rest of the paper for better readability. The dates were found to be repeating in line 180, line 190, Fig. 3, etc. A general overview of the extreme event (exact duration and maybe the range of accumulated rainfall between relevant stations) would do more justice to the third sentence in 3.1.1.

Author response: First, we have used the "7.20" storm throughout the paper. Second, we will uniformly define the duration and cumulative range of extreme event to avoid excessive repetition in the section.

- Could you provide more information on the definition on of 'early, middle, and late' period in line 185? Maybe, mention the criteria followed in Wang. et. al.

Author response: We have added the criteria adopted by Wang et. al. (2016) in line 185. In that study, storms were classified into four patterns: advanced, intermediate, delayed, and uniform depending when rainfall is most concentrated. The dimensionless durations were separated into three equal time periods. Advanced pattern, intermediate pattern and delayed pattern when more than 40% of the rainfall occurs in the first, second and third time periods, respectively. The rainfall distribution is regarded as uniform pattern otherwise. The detailed information on this has been added to the manuscript.

- line 210: 'new hourly rainfall intensity record' – could you provide the intensity value (mmh-1) in brackets if available? If rainfall amount (depth in mm) was referred to here, rephrasing so would be more accurate.

Author response: We have inserted brackets after the 'new hourly rainfall intensity record' and fill in "201.9 mm· h$^{-1}$" in the brackets.

- line 225: the last sentence feels misplaced here as no information was discussed on the vegetation of the region.

Author response: We have deleted this sentence.

- A minor suggestion: could you synthesize the results from sub-sections into one at the end of section 3.1. Since erosivity is estimated from energy, it will be worthwhile to make some general comments from the separate observations.

Author response: We have synthesized the results from sub-sections into one at the end of section 3.1 for better readability.

**(3.2) How extreme is the event recorded at Zhengzhou meteorological station?**

- 3.2.1: preposition missing in the title - page 11: line 254

Author response: We have changed 'Frequency of occurrence the maximum daily and event rainfall erosivity' to 'Frequency of occurrence of the maximum daily and event rainfall erosivity'.

- '43354' to '43,354', line 265: '58874' to '58,874'

Author response: We have changed '43354' in line 264 to '43,354', and '58874' in line 266 to '58,874', as well as other places to make the format consistent.

- Since 7.20 is suggested to be regarded as an outlier in frequency analysis (line 235), should there be additional justifications for using this for pushing envelope curves further?

Author response: This is a good point. Our frequency analysis suggests that the return period of the 7.20 storm is highly likely to be at least > 500 years, and could be as large as around 10,000 years. The envelope curve was pushed upward around the mid latitude in China because of the unusual extreme event.

- refer the points mentioned in general comments for this section

Author response: Please refer to the response on the general comments.

---

## Author Response (AR2)

**Authors' response to Editor's decision and comments from Reviewers**

**The most extreme rainfall erosivity event ever recorded in China: The "7.20" storm in Henan province**

Yuanyuan Xiao, Shuiqing Yin, Bofu Yu, Conghui Fan, Wenting Wang, Yun Xie

We would like to thank the Editor and Reviewers for their comments and suggestions. In the revised version, we have improved the text and figures, considered and addressed all the issues raised by the reviewers. We hope this revision is satisfactory for the further processing of this paper.

**Dear Editor**,

Thank you very much for your feedback. We revised the manuscript in detail, have answered all reviewers' questions. We would like to submit the revised version of our paper "The most extreme rainfall erosivity event ever recorded in China: The "7.20" storm in Henan province" by Yuanyuan Xiao, Shuiqing Yin, Bofu Yu, Conghui Fan, Wenting Wang and Yun Xie.

We believe that we followed all the suggestions from the reviewers and we have substantially improved the manuscript and we are looking forward to hearing from you with respect to the review process.

We provide responses to comments from Reviewers below. For clarity, each response is structured as follows: (1) RC# comments from Referees (black), (2) Authors' response (blue).

Thank you for your time and consideration.

Sincerely,

Yuanyuan Xiao

**RC1: 'Comment on hess-2022-351', Anonymous Referee #1, REPLY**

The authors have revised the manuscript and addressed many of the reviewers' comments. However, there are still some aspects that need to be improved:

Response: Thank you for your careful review and additional suggestions. We revised the text to address all the weaknesses that you pointed out. Below we are providing our detailed response to your specific comments in blue.

Firstly, I suggest testing and applying several KE-I equations and not just the RUSLE2 equation in order to evaluate the impact of the KE-I equation on the derived extreme rainfall erosivity values. I think that the adopted procedure for the rainfall erosivity estimation should try to cover different uncertainties related to the calculations (and measurements) and derive an estimate of the rainfall erosivity using the confidence intervals. Also, the conversion factor for the I30 is one such elements that greatly depends on the results. Are you sure that 1.489 conversion is also relevant for this specific extreme event? Can you verify this using some half-hourly data for this event? And not just report that event rainfall erosivity was 58,874 (MJ mm)/(ha*h). During your first submission you reported a return period of around 150,000 years, now you are between 10,000 and 23,100 years. Why this huge difference? If you do evaluate the impact of other steps within the rainfall erosivity calculation you should get more reasonable estimates (with the confidence intervals, ranges).

Response: While we appreciate the argument for evaluating the effect of different KE-I relationship, we believe that testing and applying several KE-I equations are unnecessary. It is true that different KE-I relationships were included in different version of the USLE, and yet more empirical localised KR-I relationship for various sites around the world (van Dijk, A. I. J. M., 2002). However, we wanted to use the equation included in RUSLE for consistence and for meaningful comparison to rainfall erosivity values reported elsewhere in China and indeed around the world (Benavidez et al., 2018; Yin et al., 2015).

Secondly, Rainfall erosivity is usually calculated based on long term precipitation records from rain gauges, and depends strongly on the temporal resolution of the precipitation data used. High-temporal resolution data at fixed intervals are increasingly available and they will to improve the estimation of rainfall erosivity, especially the event $EI_{30}$ index in the long run. Such data are still in short supply, short in in length and sparse in spatial coverage. R-factor values decrease with decreasing resolution of the precipitation data because intensity peaks are reduced when precipitation amount is aggregated over longer time intervals (Fischer et al., 2018). Therefore, it is necessary to use conversion factors to adjust the computed $EI_{30}$ value using data of low resolution. Yue et al. (2020) evaluated the impact of the time interval from 1-min to 1-hr on the 1-10-in-year $EI_{30}$ using the 1-min temporal resolution rainfall data from 62 meteorological stations in mainland China. The conversion factor for the 1-in-10-year $EI_{30}$ computed with the 60-min resolution rainfall data is 1.489, which was appropriate for evaluating extreme rainfall erosivity in this paper.

To allay the reviewer's concerned, we collected 1-min temporal resolution rainfall data from Zhengzhou Meteorological Station from 2005 to 2016, and calculated the 1-in-10-year $EI_{30}$ value to be 4054 MJ, and the ratio of 1-in-10-year $EI_{30}$ value to the 60-min temporal resolution rainfall data is 2.029. But the conversion factor for individual stations in China ranges from 1.321 to 4.601 (Fig. 1). As the equivalent 'conversion factor' for Zhengzhou is much higher than the 'average' conversion factor we used for this study, it is worthwhile to compute the standard error (± 0.064) for the conversion factor (Yue et al., 2020).

[Figure]

**Figure 1**. A comparison of 1-in-10-year EI30 values estimated with 1-min versus 60-min data. The dashed line represents the best fit using a linear model through the origin. Open circles are 1-in-10-year EI30 values estimated 60-min data without conversions, and solid circles are values adjusted with a conversion factor of 1.489 for the 62 meteorological stations in China. The open and solid circles in red refer to the Zhengzhou meteorological station. (Yue et al., 2020)

Finally, the significant difference in estimated value of the return period is related to the frequency distribution we assumed and not related to rainfall erosivity values per se. When using the LP-III, we first performed logarithmic transformation on the samples, which effectively narrowed the data range with better fit with observations. Your suggestion is valuable, it is worthwhile comparing estimates of the return period using different probability distribution explained below.

Secondly, in terms of measuring devices, for such extreme rainfall, this is import, tipping buckets have a notable decrease in accuracy with the increase of the rainfall rate. Therefore, for very extreme events, the accuracy can be a problem. Please add the specific instrument that was used, at least for the stations that was used to derive the rainfall erosivity return period. Try to include measuring device accuracy in the uncertainty estimation.

Response: The instrument used by China Meteorological Administration is SL3-1 tipping bucket rain sensors, and precipitation was measured according to the operation manual to meet certain standards at all stations. Tipping bucket rain gauges have a rainfall bearing diameter of 200 mm, and its resolution is 0.1 mm. The maximum allowable rainfall intensity is 4 mm·min$^{-1}$, and the maximum allowable rainfall error is $\pm$ 4 mm for every 100 mm. The details of the data have been added to the paper.

Thirdly, you should test if the LP-III distribution fits well to the data using some statistical test and not just say that it fits well. Moreover, as you already noted, the estimated return period greatly depends on the selected distribution function. I think that you should do the estimation using several distributions and select the one that fits the data the best.

Response: Generalized extreme value distribution model (GEV), P-III, and LP-III were used to fit the maximum event rainfall erosivity, calculated the correlation coefficient, and performed Kolmogorov-Smirnov test. The results showed that the LP-III distribution function was more suitable for this study. Please refer to the supplement for details.

There are several typos in the manuscript that should be corrected. Also, the English should be double-checked. Moreover, you should be less subjective and try to avoid statements like "so rare and freakish".

Response: We have carefully proof-read and revised the paper.

**Specific comments:**

L125: You need to provide more information about the IDW method applied.

Response: We have explained this in the revised version. (line 129-134)

Eq. 6 and Eq. 7, you should define the f(x) and F(x). It is not clear how confidence intervals were estimated.

Response: We have defined the probability density function and cumulative distribution function of the P-III distribution in the article, but we have not given a detailed calculation process of the confidence interval. The calculation process of the confidence interval is relatively complex. We use the approximate square difference using the Kite's (1975) method for parameter estimation and for evaluating the confidence interval, and we have included a reference to the method in the paper.

Figure 3: Caption should be improved.

Response: We have improved the caption, replaced the old one: 'Figure 3. Distribution of total rainfall over the study area, and rainfall mass curves for three stations with the largest rainfall totals.', with the new one 'Figure 3. A map of total rainfall over the study area, and rainfall mass curves for three stations with the largest rainfall totals.'.

Figure 4: The same for this one.

Response: We have improved the caption, replaced the old one: 'Figure 4. Spatial pattern of daily rainfall in the study area.', with the new one 'Figure 4. Spatial distribution of daily rainfall in the study area.'

Figure 6: The captions say that this is empirically fitted, you mean LP-III?

Response: The caption of Figure 6 is revised to read 'Figure 6. The logarithm of observed daily (a) and event (b) rainfall erosivity as a function of the return period assuming LP-III for Zhengzhou meteorological station'.

L225-226: I do not understand this sentence, please rephrase. Why this return period is so low, compared to others?

Response: Here we are translating the Zhengzhou meteorological station to the upper limit of the confidence interval, and found that within the range of the confidence interval, the minimum values of daily and event rainfall erosivity return periods at the Zhengzhou meteorological station are at least 785 and 516 years, respectively.

Figure 7: Confidence intervals are very wide. I am not sure how did you obtain that the estimate is between 10,000 and 23,100 years? I would expect something like: "The return period of the event was estimated to be XYX years with the 95% confidence intervals of +-YXYX years".

Response: We agree with you on this. We have explained this in the revised version.

Figure 8: Caption says this is return period, but legend is showing erosivity?

Response: We have changed the erosivity in the legend to the return period.

Figure 9: As indicated in the previous round, I do not fully understand this, you have one extreme event (point 6) and you are drawing some envelopes. You could draw several different ones.

Response: Yes, you are right. When we draw the envelopes, we do not use any equation or model, just connect the outermost points. Our goal is to emphasize that the value of rainfall erosivity for the extreme event is the highest in mid-latitude around 35 °N, which indicates the likely location of the maximum rainfall erosivity.

**References**

Benavidez, R., Jackson, B., Maxwell, D., and Norton, K.: A review of the (Revised) Universal Soil Loss Equation ((R)USLE): with a view to increasing its global applicability and improving soil loss estimates, Hydrol. Earth Syst. Sci., 22, 6059–6086, https://doi.org/10.5194/hess-22-6059-2018, 2018.

Fischer, F. K., Winterrath, T., and Auerswald, K.: Temporal-and spatial-scale and positional effects on rain erosivity derived from point-scale and contiguous rain data, Hydrol. Earth Syst. Sci., 22, 6505–6518, https://doi.org/10.5194/hess-22-6505-2018, 2018.

Kite, G. W.: Confidence limits for design events, Water Resour. Res., 11(1): 48–53, https://doi.org/10.1029/WR011i001p00048, 1975.

van Dijk, A. I. J. M., Bruijnzeel, L. A., and Rosewell, C. J.: Rainfall intensity-kinetic energy relationships: a critical literature appraisal, J. of Hydrol., 261, 1–23, https://doi.org/10.1016/S0022-1694(02)00020-3, 2002.

Yin, S. Q., Xie, Y., Liu, B. Y., and Nearing, M. A.: Rainfall erosivity estimation based on rainfall data collected over a range of temporal resolutions, Hydrol. Earth Syst. Sci., 12: 4965–4996, doi:10.5194/hessd-12-4965-2015, 2015.

Yue, T. Y., Xie, Y., Yin, S. Q., Yu, B. F., Miao, C. Y., and Wang, W. T.: Effect of time resolution of rainfall measurements on the erosivity factor in the USLE in China, ISWCR, 8, 373–382, https://doi.org/10.1016/j.iswcr.2020.06.001, 2020.

**RC2: 'Comment on hess-2022-351', Anonymous Referee #3, REPLY**

**General comment:**

The revised manuscript has incorporated the comments suggested in previous draft. The analysis has been changed from GEV to LP-III, with corresponding changes in results. Authors have improved information on devices, data, estimation of erosivity, storm classification criteria, and in summarizing the results. Some of the uncertainties are now addressed, notably the confidence of estimated return period.

**Minor comments (line no. as per latest draft):**

The language of the manuscript can be improved in some areas. For example, in line 25 'is' instead of 'as'.

Response: We have changed 'as' to 'is' in line 25, and have carefully proof-read and revised the paper.

In line 36, maybe adding 'itself' at the end of sentence could stress the idea that a very few studies have focused erositivity during event.

Response: Added.

In line 56, Zhang. et.al feels like a repetition of the more detailed descritpion in line 61.

Response: Yes, you are right. |n line 56, Zhang et al. proposed several weather systems that lead to "7.20" storm, and explained this process in detail later in line 59-64.

In line 80, were some corrections applied based on cross checking with daily observation with rain gauges? It would be worth mentioning.

Response: During quality control, some corrections have been made based on cross checking with daily observations with rain gauges.

Descriptions of model parameters and equations can be in present tense, I believe. For example, in line 98 'is' instead of 'was'. Similar in lines 115, 116, 120, 133, 142 etc.

Response: We have changed 'was' to 'is' in line 101, 118, 119, 123, 141, 151.

Since inverse distance weighing (IDW, in line 121) is used here to map rainfall erosivity distribution, the references on them could be retained (some references were there in the deleted part in introduction regarding spatial and temporal variability, line 45 in tracked changes document).

Response: We agree with you on this. We have provided additional information on IDW in Section 2.2.2 and have cited these references according to your suggestions. (line 129-134)

Figure 7 caption: it should be 'performed' in brackets, I think.

Response: We have changed 'Perform' to 'Performed' in Fig. 7.

Line 239, 240: repetition of 'by fitting the LPIII.

Response: Deleted.

Line 295: re frame the last sentence, 'following' is repeated.

Response: We have changed 'The following conclusions can be drawn as follows' to 'The following conclusions can be drawn as a result of this research'.

---

## Author Response (AR4)

**Authors' response to Editor's decision and comments from Reviewers**

The most extreme rainfall erosivity event ever recorded in China:
The "7.20" storm in Henan province

Yuanyuan Xiao, Shuiqing Yin, Bofu Yu, Conghui Fan, Wenting Wang, Yun Xie

We would like to thank the editor and reviewers for their comments and suggestions. In the revised version, we have improved the text and figures, considered and addressed all the issues raised by the reviewers. We hope this revision is satisfactory for further processing of this manuscript.

**Dear Editor**,

Thank you very much for your feedback. We have revised the manuscript in detail, and answered all the reviewers' questions. We would like to submit the revised version of our paper "The most extreme rainfall erosivity event ever recorded in China: The "7.20" storm in Henan province" by Yuanyuan Xiao, Shuiqing Yin, Bofu Yu, Conghui Fan, Wenting Wang and Yun Xie.

We believe that we have considered and addressed all the suggestions from the reviewers and have substantially improved the manuscript. We look forward to hearing from you with respect to the review process.

Below are our responses to reviewers. For clarity, each response is structured as follows: (1) RC# comments from Referees (black), (2) Authors' response (blue).

Thank you for your time and consideration.

Sincerely,

Yuanyuan Xiao

**RC1: 'Comment on hess-2022-351', Anonymous Referee #1, REPLY**

Authors have revised the manuscript and addressed most of my comments.

I suggest that authors in all parts of the manuscript (also abstract) state the possible range of erosivity (related to conversion factor), similar as done in L203-204. Also, it would be useful to add at least some details about the additional calculations done in relation to conversion factor (as shown in Figure 1 of the response to the reviewers file).

Response: Thank you for your careful review and additional suggestions. We have included the range of rainfall erosivity in the manuscript based on the range of the conversion factor. For example, for conversion factor $1.489 \pm 0.064$, the maximum event rainfall erosivity value of the "7.20" storm 58874 $MJ \cdot mm \cdot ha^{-1} \cdot h^{-1}$ is now presented as $58,874 \pm 2351$ $MJ \cdot mm \cdot ha^{-1} \cdot h^{-1}$. While we could add some details about the conversion factor, the method of calculations of the conversion factor in our response comes from Yue et al. (2020). Therefore, we have explained the range of the conversion factor based on Yue et al. (2020). We have made the following changes in the revised manuscript.

Line 125-130: Yue et al. (2020) used hourly rainfall data to calculate a conversion coefficient of 1.489 for the 1-in-10-year $EI_{30}$, which is suitable for evaluating extreme rainfall erosivity on average. The conversion factor for individual stations in China ranged from 1.321 to 4.601, and the conversion factor for Zhengzhou Meteorological Station was 2.029, higher than the average, or expected, conversion factor used for this study.   We have included the standard error $\pm 0.064$ for the conversion factor to indicate the likely uncertainty associated with this conversion factor.

As for the rain gauge instrument, was the intensity of 4 mm/min exceeded during this extreme event and how many times? Did this have an impact on the measurements and consequently on the calculations (e.g., rainfall erosivity)?

Response: The data we use is hourly rainfall data, and we do not need to calculate how many times the rainfall intensity exceeded 4 mm/min during the "7.20" storm, but we need to ensure that the hourly rainfall data are of good quality. For the rainfall intensity of 201.9 mm/h at Zhengzhou Station, this data was widely questioned when it was first released. However, according to China Daily website, the staff of Zhengzhou Meteorological Observatory use three sensors to record rainfall data, and verify the data every minute using computer calculations and manual proofreading methods to ensure that the data is not false positive and highly authentic. Therefore, we believe that we can fully use hourly rainfall data to calculate rainfall erosivity.

As for testing different KE-I equations, I still think that this would enable authors to better evaluate the possible uncertainty in the estimation of the rainfall erosivity and return periods. At least testing 2-3 more equations would be useful. But it is up to editor to decide if this needs to be included or not.

Response: There is no question that the KE-I relationship is yet another source of uncertainty as far as the total kinetic energy of the storm is concerned. We strongly advocate use of the KE-I relationship as recommended and adopted in the most recent iteration of RUSLE for reasons of standardisation and

comparison. Storm EI30 reported in this manuscript should be used and interpreted in the context of rainfall erosivity and RUSLE. That is why we believe that standard KE-I relationship is quite legitimate and adequate for the manuscript.

I hope that it is clear that most of my comments go in the direction of capturing possible sources of uncertainty in the estimation of the rainfall erosivity and return periods since obviously this event was really extreme and compared to other past events it can be regarded as some kind of outlier. Hence, statistical analysis and estimation of return periods should be done with caution and including uncertainty. Hence, authors could improve this aspect of paper.

Response: We agree with the reviewer. We used the rainfall erosivity values over the past 67 years for Zhengzhou Meteorological Station to fit the extreme value distribution fitting to estimate the return period of "7.20" storm, and the results can be uncertain because the storm was such an outlier and of the large amount of extrapolation.

**RC2: 'Comment on hess-2022-351', Anonymous Referee #3, REPLY**

**General comment:**

Previously mentioned comments have been addressed in this revised draft. However, considering the recent rain episodes in Beijing, - reported in the news as the highest in the past 142 years (as per Associate Press, August 3, 2023) – the title of the paper could be altered to account for it. Though it is beyond the scope of the paper, I would be personally interested to see if this changes any of the results.

Response: The news was misleading. In fact, the heavy rainfall in August 2023 was the largest rainfall amount recorded in Beijing over past 142 years. The heavy rainfall occurred in the Beijing area. According to the People's Government of Beijing Municipal, the maximum rainfall in the city is 744.5 mm, and the maximum hourly rainfall is 111.8 mm/h. The rainfall and rainfall intensity of August rainstorm in Beijing was nowhere as high as the "7.20" storm in Zhengzhou.

**Minor comments (line numbers on the tracked changes document):-**

Regarding the point made above, the last paragraph of the introduction where "7.20" is emphasised as 'rare' could use some rework to make the paper sound more future-proof. For example, in line 55, 'recent times' can be rephrased to 'till 2022'. I will also limit usages like 'ever'. If you are planning to study recent rain episodes in future, it can be mentioned to substantiate the importance of "7.20" in lines 73-74 etc.

Response: Good point. It is always to add a qualifier such as 'up to 2022'.

On a second reading, probably the sentence in line 36 can be removed to improve readability. For example, it could just be 'However, the long-term average value cannot fully represent….' With Diotdo et. al 2016 and Wang et. al 2022 in references.

Response: We agree with the reviewer. We deleted the sentence from line 36.

In the newly added section from lines 80 to 85, a preposition is missing in line 83. Also, it's to be specific than saying 'certain standards', 'unexpected errors' etc.

Response: We have changed 'The rainfall data acquired from CMA and the data had been quality-controlled by CMA's National Meteorological Information Center' to 'The rainfall data was acquired from CMA and the data had been quality-controlled by CMA's National Meteorological Information Center' in line 81, 'unexpected errors' to 'outlier' in line 83. We deleted the 'to meet certain standards' in line 79.

From line 115, the reasoning provided to the other reviewer on the usage of Kinnell 1981 for KE-I could be stated in the paper. Readers can also be directed to van Dijk 2002 for a review on various relationships and their use cases.

Response: We agree with the reviewer. We made the following changes in the revised manuscript.

Line 112-115:

The most widely accepted kinetic energy-intensity relationship is the exponential model proposed by Kinnell (1981). This equation has the general form:

$$e_r = e_{max} \cdot [1 - a \cdot exp(-b \cdot i_r)] \tag{1}$$

where $e_{max}$, $a$, $b$ are empirical constants. Among them, the coefficients $a$ and $e_{max}$ determine the minimum kinetic energy content. On the other hand, the coefficient $b$ defines the general shape of the curve (Kinnell, 1981).

Line 231, I think there are some mistakes in the newly added sentences, on distribution names.

Response: We deleted the redundant word 'and P-III' in line 235.

If my understanding of the recent rain episode is correct, I would recommend mentioning it in conclusions part 3 (lines 324 - 327) since its occurrence illustrates the importance of this study.

Response: Please refer to the reply in the general comment.

---

## Author Response (AR5)

**Authors' response to Editor's decision and comments from Reviewers**

**The most extreme rainfall erosivity event ever recorded in China up to 2022: The "7.20" storm in Henan province**

Yuanyuan Xiao, Shuiqing Yin, Bofu Yu, Conghui Fan, Wenting Wang, Yun Xie

We would like to thank the editor and reviewers for their comments and suggestions. In the revised version, we have improved the text and figures, considered and addressed all the issues raised by the reviewers. We hope this revision is satisfactory for further processing of this manuscript.

**Dear Editor**,

Thank you very much for your feedback. We have revised the manuscript in detail, and responded to all the reviewers' comments. We would like to submit the revised version of our manuscript "The most extreme rainfall erosivity event ever recorded in China up to 2022: The "7.20" storm in Henan province" by Yuanyuan Xiao, Shuiqing Yin, Bofu Yu, Conghui Fan, Wenting Wang and Yun Xie.

We believe that we have considered and addressed all the suggestions from the reviewers and have substantially improved the manuscript.    We look forward to hearing from you soon.

Below are our responses to reviewers. For clarity, each response is structured as follows: (1) RC# comments from Referees (black), (2) Authors' response (blue).

Thank you for your time and consideration.

Sincerely,

Yuanyuan Xiao

**RC1: 'Comment on hess-2022-351', Anonymous Referee #1, REPLY**

**Comment:** Regarding the maximum 1-minute intensity I do not agree with authors that this is not important if they used hourly data. I think this is very important since if there are errors in 1-minute data this is only aggregated to the hourly data. But if the same rainfall amount was measured using three different sensors then this is another thing that should definitely be mentioned in the manuscript. Hence, more details should be added regarding this verification using different sensors (which sensors, what was the difference, etc.).

Please, clarify this issue in the section on Materials and Methods, and if needed also in the section discussing the limitations of the study.

Response: We agree with the reviewer, Accuracy of the one-min rainfall intensity data is very important. Unfortunately, we are unable to obtain detailed information about the one-min data. In addition, we'd like to clarify that the three sensors, SL3-1, were used at each station, measurements with these sensos were used for verification purposes to ensure data quality. We have made the following changes in the revised manuscript.

Line 81-84: A multi-sensor system was used for precipitation measurement. The system consists of three separate SL3-1 tipping bucket rain sensors. Multi-sensor automatic weather stations detect abnormal or missing rainfall data caused by rain sensor failures to ensure precipitation data quality (He and Huang, 2015).

**Comment:** since you have applied only one KE-I equation, even though there are quite some presented in literature and you have not compared them, please, add this fact as a limitation to this study - also stating that when using another/different KE-I equation, the obtained results, such as the extremeness of this recorded event and therefore also its return period, may have been different as presented in this study.

Response: Different KE-I equations will lead to different results. It would be useful to acknowledge the uncertainty arising from the KE-I relationships. We have carefully compared the effects of using several KE-I equations including USLE, RUSLE, RUSLE2, van Dijk's, and found that different equations produced differences for this extreme event. We have added this as a supplement to this study, and included in the manuscript that when different KE-I equations were used, the erosivity value and return period of "7.20" storm were slightly different. At the same time, taking into account the concerns of several reviewers, we have added a discussion section in the manuscript to explain the uncertainty due to the KE-I equation, conversion factors, and frequency distributions. Please refer to the supplementary material for details. We have added the following content to the revised manuscript.

**4 Discussion**:

The above analysis shows that the "7.20" storm is the largest in terms of the rainfall erosivity among 2420 meteorological stations in mainland China up to 2022. However, there are limitations and uncertainties in our assessment due to KE-I equations, $EI_{30}$ conversion factors, and probability distributions used.

Firstly, soil erosion processes are related to rainfall kinetic energy, which is a function of the size and fall-velocity of raindrops. Different KE-I relationships were recommended in different version of the USLE, and yet more location-specific KE-I relationships were noted for various regions around the world (van Dijk et al., 2002). Using different KE-I relationships, including those for the USLE, RUSLE, and from van Dijk et al. (2002), in addition to the RUSLE2 equation adopted for the study shows that other KE-I

relationships would underestimate kinetic energy. Storm energy for the "7.20" storm using other KE-I relationships was 3.1% to 8.2% smaller than reported in the study, and the annual maximum event kinetic energy from 1951 to 2020 would differ by -16.9% to 28.7% from that reported in the study (Table S1.2). The uncertainty associated with different KE-I relationship does not increase with the magnitude of the rainfall event as shown in Fig. S1.1. Similarly, there are considerable differences in the estimated return periods of the event in terms of rainfall erosivity using different KE-I equations (Fig. S1.2). The return period of the "7.20" storm varied from about 20 thousand to more than 50 thousand years. The relatively small difference in event KE can lead to considerable differences in the return period for such an extreme event when the KE value of event exceeded all other KE values for the site by at least an order of magnitude. These large uncertainties associated with the return period of extreme precipitation have been noted in Germany (Grieser et al., 2007).

Secondly, rainfall erosivity is usually calculated using long term precipitation records from rain gauges, and depends strongly on the temporal resolution of the precipitation data used. Data at higher temporal resolution would be higher desirable to compute rainfall erosivity is high temporal resolution. However, such data are in short supply, short in length and sparse in spatial coverage. R-factor values decrease with decreasing temporal resolution because intensities are reduced when precipitation amount is aggregated over longer time intervals (Fischer et al., 2018). Therefore, it is necessary to use conversion factors to adjust the computed $EI_{30}$ value using data of low temporal resolution. The conversion factor for the 1-in-10-year $EI_{30}$ computed with the one-min resolution rainfall data is 1.489, which was appropriate for evaluating extreme rainfall erosivity in this study. To allay the reviewer's concerned, we collected one-min temporal resolution rainfall data from Zhengzhou Meteorological Station from 2005 to 2016, The annual maximum $EI_{30}$ values estimated using one-min and one-hour data were compared (Fig. S2.2). The conversion factor for the annual maximum $EI_{30}$ at Zhengzhou meteorological station is 1.974, which is very close to the conversion factor of 1-in-10-year $EI_{30}$ is 2.029.

Finally, the estimated return period depends on the selected probability distribution function. Different probability distribution functions can produce quite different estimates for large return periods (Laio et al., 2011). Three frequency distributions were considered and tested, including Generalized extreme value (GEV), P-III, and LP-III was found to be the most appropriate (Table S3.1). All the three distributions fitted the observations well, and performance indicator values do not suggest a single distribution that consistently and significantly superior to others (Table S3.2). The return period estimated by the three probability distributions are quite different. The average recurrence intervals of the maximum event rainfall erosivity of GEV and P-III for "7.20" storm exceeds 340,600 years, which is far greater than reported in the study. The estimated return period of around 20,000 years for the "7.20" storm is conservative. The estimated return period would be much higher if we use other KE-I equations and other probability distributions. Given LP-III was widely recommended for extreme precipitation and flood events in China (Chen et al., 2012), LP-III was used to assess the return period of this "7.20" storm. Estimating return periods comes with large uncertainties, especially for return periods exceeding the length of the observational record (Bloemendaal et al., 2020).

**RC2: 'Comment on hess-2022-351', Anonymous Referee #2, REPLY**

**Comment:** the title and the text are using the expression "the most extreme rainfall erosivity event ever recorded in China" - which is fine, but you must the date, e.g. till 2022, so that possibly more extreme

events in future will be regarded as a new extreme - this is quite easy to incorporate into the title and the text.

Response: We have changed the title "The most extreme rainfall erosivity event ever recorded in China: The "7.20" storm in Henan province" to "The most extreme rainfall erosivity event ever recorded in China **up to 2022**: The "7.20" storm in Henan province". In addition, we have added "up to 2022" in the following position of the manuscript.

Line 20: "and these were the maximum rainfall erosivity ever recorded among 2420 meteorological stations in mainland China **up to 2022**"

Line 65: "The maximum hourly rainfall between 16:00 and 17:00 on 20 July reached 201.9 mm at Zhengzhou meteorological station, the highest ever recorded in China **up to 2022**"

Line 265: "geographical distribution of the maximum daily rainfall erosivity ever recorded at each of 2420 meteorological stations in China **up to 2022** is shown as a function of the latitude in Fig. 9."

Line 279: "Geographical distribution of the maximum event rainfall erosivity ever recorded at each of 2420 meteorological stations in China **up to 2022** is shown as a function of the latitude in Fig. 10."

In addition, we have revised and updated the manuscript because we found that there was a calculation error in the rainfall erosivity used for frequency analysis. Using a conversion factor of 1.489 consistently, we updated the relevant results in the manuscript.